# HSP90 differentially stabilizes plant ABCB-type auxin transporters on the plasma membrane

Tashi Tsering [1,7], Martin Di Donato[1,7], Despina Samakovli[2], Dimitra Milioni[2], Francesca Romana Iacobini [1], Konstantinos Panagiotopoulos [2], Panagiota-Konstantinia Plitsi[2], Elisa Azarello[3], Stefano Mancuso[3], Vendula Pukyšová[4], Marta Zwiewka[4], Tomasz Nodzynski [4], Michael Stumpe[1], Jutta Ludwig-Müller [5], Aurélien Bailly [6], Polydefkis Hatzopoulos [2] ✉ & Markus M. Geisler [1] ✉

Closely related FKBP orthologs, FKBP42/TWISTED DWARF1 (TWD1) and FKBP38, have been shown to control the biogenesis of plant and mammalian ATP-binding cassette (ABC) transporters, respectively. However, the mechanistic role of the described FKBP-ABCB interaction is widely unknown. Here, we verify cytosolic HEAT-SHOCK PROTEIN90 (HSP90) isoforms as valid interactors of TWD1 and map HSP90 binding to an amphiphilic alpha-helix preceding its TPR domain. We provide pharmacological and genetic evidence that a subset of TWD1-interacting ABCBs, in contrast to mammalian ABCBs, are constitutive HSP90 clients in plants. This effect and its specificity are presumably provided by TWD1. Our data strongly correlate the impact of HSP90 inhibition on ABCB-mediated development and ABCB plasma membrane stability on the one hand and ABCB cycling rate on the other. In summary, we uncover a dynamic mechanism of HSP90 for differential stabilization of the plasma membrane ABCB isoforms to regulate polar auxin transport and to confer developmental plasticity.

Current work has provided evidence that the immunophilin-like FKBP42, TWD1, functions as a co-chaperone of a subset of B-type ABC transporters (ABCBs)[1–3], which are essential for the cell-to-cell movement of the plant hormone, auxin[4–7]. During this process, referred to as polar auxin transport (PAT), auxin is gradually and directionally redistributed within tissues creating local auxin minima and maxima essential for plant performance and development[8,9]. Establishment, maintenance, and fine-tuning of auxin gradient is thought to be mainly provided by the independent and combined action of ABCB- and PIN-FORMED (PIN)-type auxin exporters[3,9–11]. Recently,

Arabidopsis ABCB1 and ABCB19 were shown to transport along with auxin, brassinosteroids (BR), another class of structurally different growth-promoting hormones[12–14]. This enhanced specificity of plant ABCBs for a few substrates was recently referred to as substrate multispecificity[15].

In the *twd1* mutant of the model plant *Arabidopsis thaliana*, plasma membrane-localized ABCB1,4,19 are extensively retained on the ER leading to their subsequent degradation and reduction of auxin transport and plant growth[1,2]. This suggests defects in early ABCB biogenesis in *twd1* and in agreement, *twd1* and *abcb1,19* mutants show

[1]University of Fribourg, Department of Biology, Fribourg, Switzerland. [2]Agricultural University of Athens, School of Applied Biology and Biotechnology, Department of Biotechnology, Athens, Greece. [3]LINV-DIPSAA, Università di Firenze, Florence, Italy. [4]Mendel Centre for Plant Genomics and Proteomics Masaryk University, CEITEC MU, Brno, Czech Republic. [5]Technische Universität Dresden, Faculty of Biology, Dresden, Germany. [6]Department of Plant and Microbial Biology, University of Zürich, Zürich, Switzerland. [7]These authors contributed equally: Tashi Tsering, Martin Di Donato. ✉e-mail: phat@aua.gr; markus.geisler@unifr.ch

widely overlapping dwarf phenotypes and disoriented ("twisted") cellular and organ growth[1,2,16,17]. In analogy, the mammalian ortholog of FKBP42/TWD1, FKBP38, associates with and controls steady-state levels of the voltage-dependent, delayed rectifier potassium channel, HERG (HUMAN ETHER-A-GO-GO; ref. 18), and CFTR (CYSTIC FIBROSIS TRANSMEMBRANE CONDUCTANCE REGULATOR)/ABCC7[7,19,20]. CFTR is a C-/MRP-type ABC transporter, functioning as a chloride channel whose mutation is responsible for the genetic disease mucoviscidosis or cystic fibrosis[21]. CFTR folding relies on the FKBP38 *cis-trans* peptidyl-prolyl isomerase (PPIase) function, an activity that is thought to be negatively regulated by HSP90[22]. However, all attempts to validate a PPIase activity on TWD1 have failed[3] leaving the mechanistic role of described TWD1-ABCB[1,2,16,17] interaction an open issue.

Previous work suggested that TWD1 functions together with HEAT SHOCK PROTEIN 90 (HSP90) as chaperones for a subset of ABCBs[3,7,16,17,23]. *Arabidopsis* encodes seven HSP90 isoforms, four of which (HSP90.1-4) are cytoplasmic. HSP90.5 is localized to the chloroplast, HSP90.6 to the mitochondria and HSP90.7 (also called SHEPHERD (SHD)) to the ER lumen[24]. Recently, receptors and signaling components for several plant hormones have been identified as clients of the HSP90 chaperone system[24–26], suggesting a direct HSP90-dependent link to auxin, jasmonic acid (JA) and BR signaling. While a defect in hypocotyl elongation in the *tir1-1* mutant at elevated temperatures (28 °C) has been noted early on, quite recently HSP90 and cognate co-chaperones were shown to stabilize TIR1 and AFB2 upon increasing temperature[27]. Besides hormone signaling, HSP90 was recently shown to directly regulate PAT: Depletion of HSP90 affected the asymmetrical distribution of PIN1 and the polar distribution of auxin, resulting in impaired root gravitropism and lateral root formation[28].

Lately, Arabidopsis HSP90 isoforms 1 and 3 were found to be part of the TWD1 interactome[29] and Arabidopsis HSP90.1 was found to bind TWD1 with a low micro-molar $K_D$[30]. The TPR domains of FKBPs consist of at least three loosely conserved 34-amino acid TPR motifs that bind to the MEEVD peptide of the HSP90 C-terminus[24]. As HSP90s exhibit an ATPase-dependent chaperone activity, HSP90 specificity to a sub-class of their client proteins can be provided by the FKBPs, which function as HSP90 co-chaperones[7,31]. While the precise mechanism of ABCB regulation by TWD1 is still not resolved, it was speculated to have some degree of analogy to the regulation of human ABCC7/CFTR[7]. Intriguingly, CFTR is a HSP90 client[7,20,32] and its regulation involves the multifunctional immunophilin FKBP38, which has a homologous domain structure to TWD1[7,20].

## Results

### HSP90 binds to helix 7 of TWD1

In this study, we address the impact of cytoplasmic HSP90 chaperones on the differential stabilization of ABCB-type auxin transporters at the PM that is orchestrated by the co-chaperone TWD1. To consolidate previous TWD1-HSP90 interaction data, we analyzed public-available expression data of *TWD1* and *HSP90*[33] indicating strong co-expression in the root tip between *TWD1* and *HSP90.1* and *HSP90.3* but less with *HSP90.2* or *HSP90.4* (Supplementary Fig. 1a). In silico data were validated by confocal analyses of crosses of *HSP90:HSP90.1-mNeonGreen* and *TWD1:TWD1-CFP* lines indicating nearly perfect colocalization (Supplementary Fig. 1b). Next, we co-transformed tobacco with *TWD1-RFP* or *free RFP* and *YFP-HSP90.3* by employing agrobacterium-mediated leaf infiltration. Fluorescence lifetime (FRET-FLIM) imaging (Fig. 1a) and co-immunoprecipitation (co-IP) data (Fig. 1b) determined that YFP-HSP90.3 co-localizes (Supplementary Fig. 1f) and interacts with TWD1-RFP but not with free RFP used here as a negative control. Surprisingly, mutating putative key residues in the TPR domain of TWD1 (N187A and K265A; Supplementary Fig. 1c) did not significantly impair TWD1-HSP90.3 interaction, as revealed by FRET-FLIM (Fig. 1a)

and co-IP (K265A; Fig. 1b). Using the recently resolved structure of the human HSP90:FKBP51:p23 complex, which identified a helix extension (helix 7) as a critical HSP90 recognition motif[34], we modeled putative TWD1-HSP90 contact sites (Supplementary Fig. 1c). Based on FRET-FLIM, mutation of F326 (equivalent to FKBP51 F413[34]) completely disrupted TWD1-HSP90.3 interaction (Fig. 1a), while co-IP showed a reduction compared to Wt and TWD1$^{K265A}$ (Fig. 1b). The observed differences employing these techniques might be attributed to the fact that point mutation can lead to altered distances between proteins that are detected by FRET, while substitution of single amino acids might have less effect on co-IPs as protein–protein interactions employ entire surface domains. In summary, these data imply that in analogy to the FKBP51-HSP90 complex, TWD1-HSP90 interaction is mainly provided by helix 7, which was originally thought to be involved in calmodulin binding[30]. Importantly, none of the TPR domain and helix 7 mutations significantly affected the ability of TWD1 to promote ABCB1-mediated auxin (IAA) export (Fig. 1c; Supplementary Fig. 1c; ref. 35). Co-expression did not significantly alter the interaction between ABCB1 and TWD1 as exposed by FRET (Fig. 1d) and BRET analyses (Supplementary Fig. 1g), or the expression of ABCB1 (Supplementary Fig. 1d). Taken together, this indicates that HSP90 binding to helix 7 has no direct effect on the upregulation of ABCB1-mediated auxin transport by TWD1, which is thought to be provided by the PPIase of the TWD1 FKBD[3,7].

In order to dissect the role of HSP90 binding to helix 7 of TWD1 in either early biogenesis or PM stabilization of mature ABCBs, we complemented *twd1-3* lines expressing *ABCB1:ABCB1-GFP*[2] with Wt, TPR (*TWD1$^{K265A}$*) and helix 7-mutated versions (*TWD1$^{F326K}$*) of *TWD1*. Like Wt *TWD1*, *TWD1$^{K265A}$*, and *TWD1$^{F326K}$* nearly fully restored the overall growth defects of *twd1* (Fig. 1e–g), exemplified by complementation of its stunted growth (Fig. 1e) and reduced root lengths (Fig. 1f, g; ref. 17). However, we noticed that root lengths in *TWD1$^{K265A}$* and *TWD1$^{F326K}$* lines showed a wider distribution compared to Wt *TWD1* (Fig. 1f, g). Overall complementation is in line with the finding that ABCB1 is, unlike in *twd1* (Fig. 1i), able to reach the PM in all complemented lines (Fig. 1h, i). However, quantification of ABCB1 signals revealed reduced PM presence in *TWD1$^{F326K}$* and *TWD1$^{K265A}$* compared to Wt *TWD1* lines, which was not statistically different from *twd1* (Fig. 1i, j). Additionally, for the helix 7 mutation but not for *TWD1$^{K265A}$* we found aberrant cell divisions in the root tip (Fig. 1i) as previously found for *HSP90$^{RNAi}$* lines[28], verifying on one hand that F326K mutation significantly affected TWD1 binding to HSP90 and on the other that this event is critical for correct cell division.

Together, these data establish a physical interaction between TWD1 and cytoplasmic HSP90 isoforms to helix 7 of TWD1 but exclude a major role of TWD1-HSP90 interaction in early ABCB biogenesis. Instead, they indicate that HSP90 activity is a critical factor for the PM presence of mature ABCBs.

### Auxin-transporting ABCBs that interact with and are regulated by TWD1 are HSP90 clients

The absence of defects in early ABCB1 biogenesis by disruption of TWD1-HSP90 interaction encouraged us to investigate the impact of HSP90 on the expression of mature ABCB1, meaning its PM presence. Cytoplasmic HSP90 isoforms were shown to share closely redundant functions, while higher order *HSP90* loss-of-function mutants were found synthetically lethal[36]. Therefore, to test the effect of HSP90 inhibition on ABCB expression and localization we employed the highly-specific HSP90 inhibitor, geldanamycin (GDA)[37]. Low micro-molar concentrations of GDA dramatically reduced ABCB1 at the PM and increased accumulation in the cells of the root tip (Fig. 2a, b; Supplementary Fig. 2a–e). Western blot and confocal analyses revealed that in the presence of cycloheximide, total ABCB1 protein abundance was significantly decreased 2 h after GDA treatment (Supplementary Fig. 2a–c). This finding is in line with the general concept that HSP90

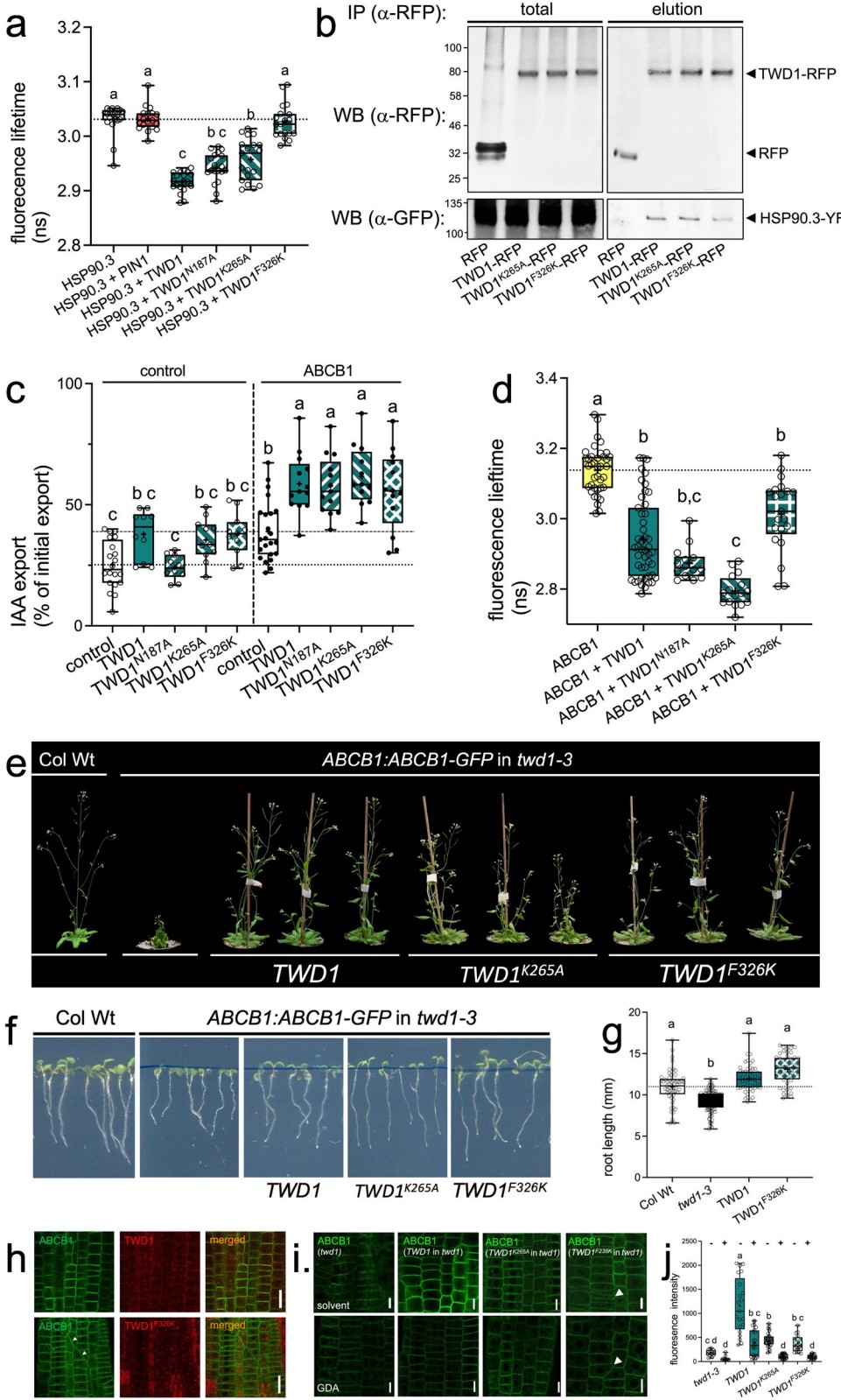

acts as a chaperone of mature proteins[38]. ABCB1 degradation by HSP90 inhibition is specific to ABCB-type auxin transporters because the total abundance of PIN2 auxin efflux carrier or the permease-like auxin importer AUX1, both belonging to evolutionary distinct, secondary active auxin transporter classes[39], were not significantly affected by GDA treatment (Fig. 2a, b). The same was found for TWD1 itself (Fig. 2a, b) or the unrelated PM protein aquaporin, PIP2A, or the vacuolar or ER

markers, V-PPase and BiP, respectively (Supplementary Fig. 2d). Based on Western blotting using an anti-HSP90 antiserum, CHX seems to up-regulate HSP90 expression (Supplementary Fig. 2a, b). This is unex-pected but CHX might simply cause general stress to the plant that results in this HSP90 upregulation.

ABCB1 degradation was accompanied by a shift from the PM to vacuolar membranes (Fig. 2a; Supplementary Fig. 2d), constituting

**Fig. 1 | TWD1 is a co-chaperone of HSP90 binding to helix 7 of TWD1. a** Mutation of F326 in helix 7 of TWD1 disrupts TWD1-HSP90.3 interaction analyzed by FRET-FLIM after co-expression of indicated Wt and mutated versions of *35S:TWD1-RFP* and *35S:YFP-HSP90.3* in tobacco. Significant differences ($p < 0.05$) of means ± SE ($n = 3$ independent tobacco transfections, each with 6–8 cells) were determined using Ordinary One-way ANOVA (Tukey's multiple comparison test) and are indicated by different lowercase letters. **b** Co-immunoprecipitation of HSP90.1 and HSP90.3 after co-transfection of *35S:TWD1-RFP* or *35S:RFP* with *35S:YFP-HSP90.3* in tobacco. IP was performed using anti-RFP columns, Western detection using anti-RFP and anti-GFP. $n = 3$ independent tobacco transfections and co-IPs; an exemplary result is shown. **c** TPR-HSP90 interaction is not essential for ABCB1 activation by TWD1 analyzed by protoplast transport assays after co-expression of indicated Wt and mutated versions of *35S:TWD1-YFP* and *35S:RFP-HSP90.3* in tobacco. Significant differences ($p < 0.05$) of means ± SE ($n = 12$–18 independent tobacco transfections and protoplast preparations) were determined using Ordinary One-way ANOVA (Tukey's multiple comparison test) and are indicated by different lowercase letters; diffusion control in Supplementary Fig. 1h. **d** Mutation of F326 does not interfere with TWD1-ABCB1 interaction analyzed by FRET-FLIM after co-expression of indicated Wt and mutated versions of *35S:ABCB1-YFP* and *35S:TWD1-mCherry* in tobacco. Significant differences ($p < 0.05$) of means ± SE ($n = 3$ independent tobacco transfections with 6–8 cells) were determined using Ordinary One-way ANOVA (Tukey's multiple comparison test) and are indicated by different lowercase letters. **e–g** Phenotypic complementation of *ABCB:ABCB1-GFP* (in *twd1-3*) with Wt and a helix 7 mutation (TWD1[F326K]) of *35S:TWD1-mCherry*; phenotypes of soil-grown (**e**) and root lengths of plate-grown plants (**f**, **g**). Significant differences ($p < 0.05$) of means ± SE ($n = 3$ independent experiments with each 10 seedlings) were determined using Ordinary One-way ANOVA (Tukey's multiple comparison test) and are indicated by different lowercase letters. **h** ABCB1-GFP and TWD1-mCherry imaging of *ABCB:ABCB1-GFP* (in *twd1-3*) lines complemented with Wt and a helix 7 mutation (TWD1[F326K]) of *35S:TWD1-mCherry*. ABCB1-GFP in Wt and *twd1-3* is shown as a reference. Arrowheads mark aberrant cell divisions in the root tip; bars, 100 μm. **i, j** Confocal imaging (**i**) and quantification (**j**) of ABCB1-GFP PM signals of *ABCB:ABCB1-GFP* (in *twd1-3*) lines complemented with Wt (TWD1), TPR (TWD1[K265A]) and helix 7 mutation (TWD1[F326K]) of *35S:TWD1-mCherry* treated with 5 μM GDA (+) or solvent control (−). Note that imaging of ABCB1 was performed under equal excitation conditions allowing for direct comparison. Arrowheads mark aberrant cell divisions; bars, 50 μm. Significant differences ($p < 0.05$) of means ± SE ($n = 3$ independent GDA treatments with 18–20 cells) were determined using Ordinary One-way ANOVA (Tukey's multiple comparison test) and are indicated by different lowercase letters. Data are presented as box-and-whisker plots, where median and 25th and 75th percentiles are represented by the box itself and the middle line, respectively; means are indicated by a "+". Source data are provided as a Source data file.

organelles of protein degradation in plant cells. After exposure to FM4-64 for 3 h, the membrane marker FM4-64 surrounds intra-vacuolar GFP signal caused by ABCB1-GFP degradation, while GDA treatment results in colocalization of FM4-64 and GFP signals on vacuolar membranes (Supplementary Fig. 2e, f). PM removal and vacuolar degradation were enhanced by a 12 h dark treatment, resulting in simultaneous luminal vacuolar accumulation of soluble GFP-signal (Supplementary Fig. 2e, f). Additionally, we used Concanamycin A (ConcA), a specific inhibitor of the vacuolar H⁺-ATPases to reduce the acidification of lytic compartments and thus protein degradation[40]. Under our experimental conditions, we noticed different sensitivities of the ABCBs to ConcA with ABCB19 and ABCB1 being the most sensitive ones (Supplementary Fig. 2j) and responding with increased abundance at the PM upon ConcA treatment indicating aberrant endomembrane trafficking (Supplementary Fig. 2j). Contrary, ConcA decreased the presence of ABCB4 at the PM (Supplementary Fig. 2j). Notably, ConcA application in GDA-treated seedlings did not change the PM levels of ABCB1 and ABCB19 when compared to GDA-treated plants (Supplementary Fig. 2j). Next, we used wortmannin (Wort), a specific phosphatidylinositol 3-kinase (PI3K) inhibitor, which inhibits plasma membrane protein and receptor sorting and/or vesicle budding required for the delivery of endocytosed material to "mixing" endosomes[41]. Wort induced the internalization of ABCB1 and ABCB19 showing impaired vacuolar trafficking indicating that ABCB1 and ABCB19 require PI3K signaling for their translocation to lytic vacuoles, while ABCB4 was found to be insensitive to Wort treatment (Supplementary Fig. 2j). We observed mistargeting of ABCB1 and ABCB19 to the tonoplast suggesting sorting defects at the level of multivesicular body (MVB)/PVC (Supplementary Fig. 2j). In GDA-treated plants, Wort seems to inhibit early stages of endocytosis of all tested ABCB transporters at the plasma membrane (Supplementary Fig. 2j). Conclusively all three inhibitors (GDA, ConcA and Wort) affected their sorting and trafficking towards lytic vacuoles of ABCB1 and ABCB19 and had a minor effect on the ABCB4.

The expression of ABC transporters is often regulated by their own substrates, and recently, ABCB1 and ABCB19 PM expression were found to be reduced upon BL treatments[12]. Therefore, we quantified the PM presence of ABCB1, ABCB4, and ABCB19 in solvent and GDA-treated seedlings after the application of IAA (1μM for 2 h) and BL (100 nM for 4 h) in entire roots and in the root transition zone (Supplementary Fig. 4). Results indicate that all three ABCBs are strongly upregulated by IAA, while the upregulation of ABCB19 and ABCB4 by IAA was stronger in comparison to ABCB1 (Supplementary Fig. 4). Interestingly, only ABCB19 is apparently upregulated by BL; ABCB4 is downregulated, while ABCB1 was widely unaffected by BL (Supplementary Fig. 4). Importantly, hormone treatment had no significant impact on PM protein levels of ABCB1 and ABCB19 when HSP90 function was inhibited by GDA, indicating that IAA or BL responses require HSP90.

Next, we tested the impact of genetic *HSP90* reduction on ABCB1 localization by using *HSP90* RNAi lines recently reported to affect Arabidopsis root growth caused by altered PIN1 polarity[28]. Analysis of the *HSP90*[RNAi] line 10H revealed in analogy to GDA treatments, specific ABCB1 PM degradation, while expression of PIN2 and AUX1 was not significantly altered (Fig. 2c, d). Interestingly, we found a small but significant upregulation of total TWD1 abundance in *HSP90*[RNAi] line 10H (Fig. 2c, d), which is currently under investigation. Finally, despite substantial functional redundancy of HSP90 isoforms[24], we uncovered reduced PM localization of ABCB1 in the *hsp90.1* loss-of-function mutant (Supplementary Fig. 2g). In contrast to GDA treatment, the reduced PM localization of ABCB1 in both *hsp90.1* mutants and *HSP90*[RNAi] lines occurred without vacuolar redistribution, implying isoform-specific regulation of ABCB1 trafficking by other HSP90 isoforms.

The finding that pharmacological and genetic reduction of HSP90 activity resulted in ABCB1 PM destabilization suggests that ABCB1 itself is an HSP90 client. To challenge this status, we tested ABCB1-HSP90 interaction by co-IP and FRET upon co-expression in tobacco. ABCB1-YFP was able to pull down both HSP90.1 and HSP90.3 isoforms (Fig. 2e) but not AUX1-YFP, shown before to be GDA-insensitive (Fig. 2a). Co-IP data were corroborated by FRET-FLIM experiments indicating substantial loss of ABCB1-GFP fluorescence lifetime with HSP90.3-RFP but not with unrelated receptor kinase, QSK1-RFP[42] (Fig. 2f). This indicates that exactly the two HSP90 isoforms interacting with TWD1[29] are also in physical contact with ABCB1.

The Arabidopsis genome contains 22 full-size ABCBs from which 11 are considered auxin-transporting ABCBs (ATAs; Supplementary Fig. 2i) based on the presence of a signature D/E-P motif and transport studies[7,43,44]. In order to explore if all putative ATAs are HSP90 clients, we GDA-treated lines expressing fluorescent protein-tagged versions of ABCB19 and ABCB4 (shown before to interact with TWD1[1,2,29]), ABCB6 (as representative of the ABCB6,20 group 1A[45]), and ABCB18 (as representative of the ABCB15-22 clade1B[44]; Supplementary Fig. 2i). Our analysis indicated that only ABCB1,4,19 were significantly sensitive to

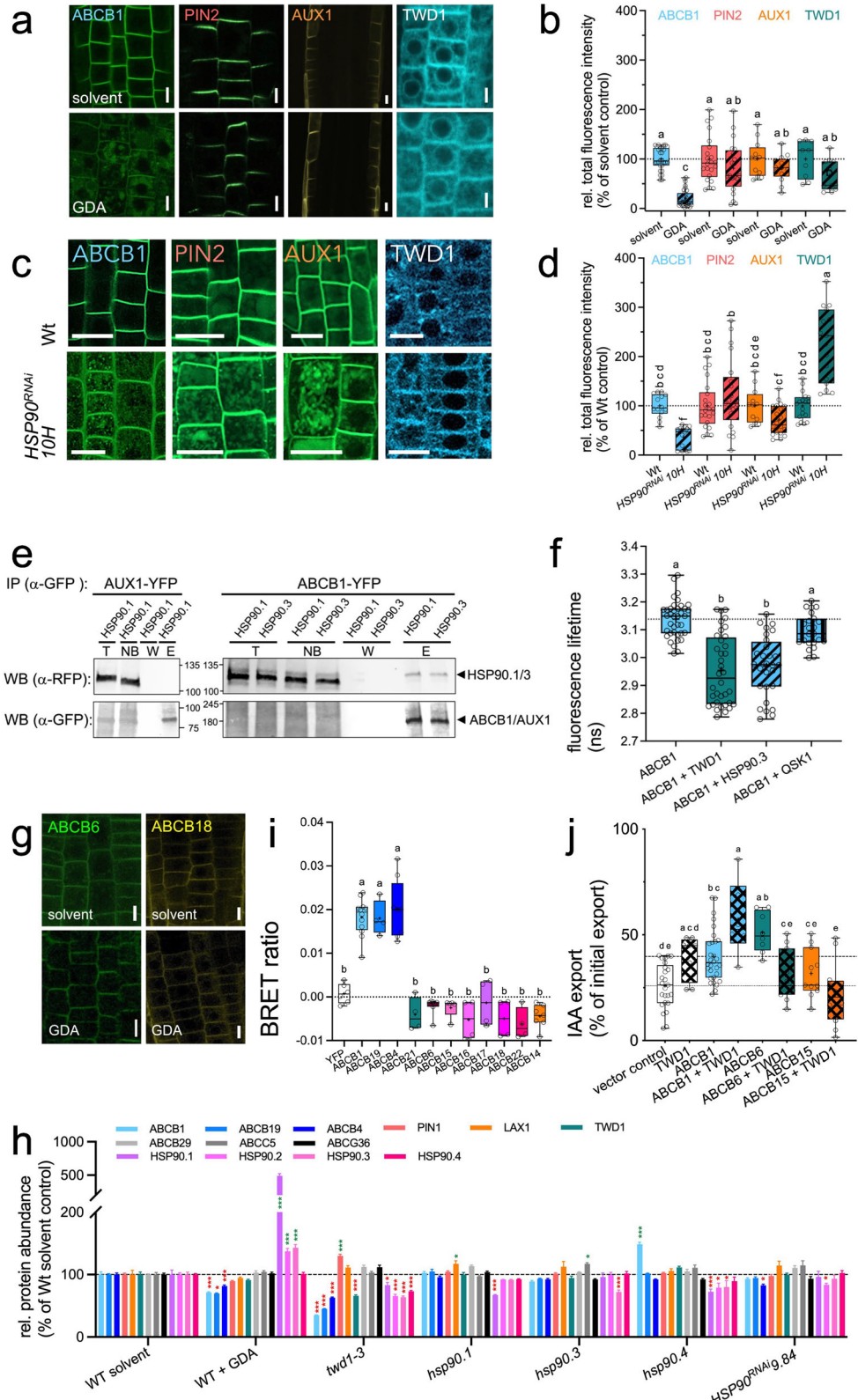

GDA as revealed by confocal imaging (Figs. 2g, 4a, b). These data were supported by TMT-based multiplexing proteomics of WT treated with GDA and of *hsp90* and *twd1* mutant lines analyzing the protein abundance of the entire Arabidopsis membrane proteome (Fig. 2h; Supplementary Fig. 3; Supplementary Data 1). In agreement with cell biological data, GDA significantly reduced ABCB1,19,4 abundance, while other auxin transporters, like PIN1 and LAX1 (AUX1 was not

detected) or other ABC transporters, like ABCB29, ABCC5, or ABCG36, were only mildly affected (Fig. 2h; Supplementary Data 1). In contrast to imaging, our proteomics analyses revealed a lower degree of ABCB1,19,4 downregulation by HSP90 inhibition or deletion, which might be due to an underestimation of actual protein/peptide abundance differences, a phenomenon known as "ratio compression"[46]. Interestingly, we observed the same overall downregulation pattern of

**Fig. 2 | Auxin-transporting ABCBs that interact with and are regulated by TWD1 are HSP90 clients. a**, **b** Confocal imaging (**a**) of GDA treated (5 μM) ABCB1-GFP, PIN2-GFP, AUX1-YFP, and TWD1-CFP in the root tip; bars, 20 μm. Quantification of total fluorescence signals corresponding to whole-region analysis measurements of the mean intensity across complete meristematic or transition zones was performed using the Zen blue 2012 software package (**b**). Significant differences ($p < 0.05$) of means ± SE ($n = 3$ independent GDA treatments with 14–20 cells) were determined using Ordinary One-way ANOVA (followed by Šídák's multiple comparisons test) and are indicated by different lowercase letters. **c**, **d** Confocal imaging (**c**) of ABCB1-GFP, PIN2-GFP, AUX1-YFP and TWD1-CFP in the root tip of Col-0 Wt (Wt) and *HSP90* RNAi line 10H (*HSP90^RNAi* 10H); bars, 100 μm. Quantification of total fluorescence signals corresponding to whole-region analysis measurements of the mean intensity across complete meristematic or transition zones was performed using the Zen blue 2012 software package (**d**). Significant differences ($p < 0.05$) of means ± SE ($n = 4$ independent experiments with 14–20 cells) were determined using Ordinary One-way ANOVA (followed by Šídák's multiple comparisons test) and are indicated by different lowercase letters. **e** ABCB1-YFP but not AUX1-YFP is able to pull-down HSP90.1 and HSP90.3 upon co-expression of *35S:ABCB1-YFP* or *35S:AUX1-YFP* and *35S:RFP-HSP90.1* or *35S:RFP-HSP90.3* in tobacco. $n = 3$ independent tobacco transfections and co-IPs; an exemplary result is shown. **f** ABCB1 interacts with HSP90.3 in planta analyzed by FRET-FLIM upon co-expression of *35S:ABCB1-YFP* and *35S:RFP-HSP90.3* in tobacco; *35S:TWD1-RFP* and *35S:QSK1-RFP* were used as positive and negative controls, respectively. Significant differences ($p < 0.05$) of means ± SE ($n = 3$ independent tobacco transfections with 10–12 cells) were determined using Ordinary One-way ANOVA (Tukey's multiple

comparison test) and are indicated by different lowercase letters. **g** GDA treatment (5 μM) does not significantly alter ABCG6 or ABCB18 expression in the root tip; bars, 50 μm. **h** Relative mean abundance of ABCB1,4,19 and selected transporters and regulatory proteins relevant for this study quantified by TMT-based 16-channel multiplexing proteomics of Arabidopsis Wt (Col Wt), *twd1* and indicated *hsp90* loss-of-function and *HSP90^RNAi* mutant microsomes. In some cases, Arabidopsis cultures were treated with Geldanamycin (GDA; 5 μM) for 24 h; $n = 4$. Significant differences ($p < 0.05$) of means ± SE ($n = 4$ independent experiments) were determined using Two-way ANOVA followed by Tukey's multiple comparisons test. Significantly upregulated and downregulated proteins compared to Wt solvent control are indicated by green and red stars, respectively (*$p < 0.05$; **$p < 0.01$; ***$p < 0.001$). **i** ATAs that are GDA-sensitive interact with TWD1 as shown by BRET analyses after co-expression in tobacco. Significant differences ($p < 0.05$) of means ± SE ($n = 4$ independent transfections) were determined using Ordinary One-way ANOVA (Dunnett's multiple comparisons test) and are indicated by different lowercase letters; colocalization control in Supplementary Fig. 2h. **j** Auxin (IAA) export capacities analyzed by protoplast transport assays after co-expression of 35S:ABCB1-YFP, 35S:ABCB6-YFP, and 35S:ABCB15-YFP with and without 35S:TWD1-RFP in tobacco. Significant differences ($p < 0.05$) of means ± SE ($n = 6$–20 independent tobacco transfections and protoplast preparations) were determined using Ordinary One-way ANOVA (Holm-Šídák's multiple comparisons test) and are indicated by different lowercase letters. Data are presented as box-and-whisker plots, where median and 25th and 75th percentiles are represented by the box itself and the middle line, respectively; means are indicated by a "+". Source data are provided as a Source data file.

ABCB1,19,4 in *twd1*. Although to a lesser extent, a similar effect was detected in both *hsp90.3* and *HSP90^RNAi9.84* line. Another exciting finding was that HSP90 was upregulated in GDA-treated WT, which might represent a compensatory mechanism[47], but downregulated in *twd1*, verifying TWD1 as a major partner of HSP90 in the membrane stabilization of ABCB1,19,4. Strikingly, our imaging (Fig. 2a–d, g) and proteomics analyses (Fig. 2h; Supplementary Data 1) revealed that the HSP90 client status of these three ABCBs correlated perfectly with the ability to interact with TWD1 as shown by BRET (Fig. 2i) or yeast two-hybrid analysis[17]. On the other hand, quantification of IAA export of representative ATAs, ABCB1, ABCB6, and ABCB15, upon co-expression with TWD1 in tobacco indicated that only TWD1-interacting ATAs were activated by TWD1 (Fig. 2j).

In conclusion, our data indicate that Arabidopsis ABCB1,4,19 represents a subset of three ATAs as HSP90 clients, thereby revealing a surprising kingdom-specific difference in HSP90 client recruitment. In contrast to these three ATAs, the PM presence of human ABCB1/PGP1 (as well as ABCC1/MRP1), both contributing to multidrug resistance phenomena in cancer cells[19], was shown to be insensitive to GDA[32]. Importantly, in analogy to human CFTR/ABCC7, which is regulated by FKBP38[7,20], the client status of plant ATAs seems to be defined by the functional interaction with FKBP42/TWD1, serving as a co-chaperone of this ABC-FKBP-HSP90 dynamic module.

## HSP90 acts as a positive regulator of ABCB-mediated polar auxin transport

Recently, pharmacological and genetic downregulation of HSP90 was shown to cause defects in auxin homeostasis and auxin-controlled root development[24,28,48], however, transport defects were not quantified. By using an external, auxin-specific micro-electrode[29,49], we were able to uncover that 5 μM GDA efficiently blocked polar auxin transport in the root tip (Fig. 3a; Supplementary Fig. 5a, d). HSP90 inhibition by GDA (66% inhibition of maximal IAA influx) was as efficient as using equal concentrations of 1-*N*-naphtylphtalamic acid (NPA; 65% inhibition[50]), an established non-competitive auxin efflux inhibitor shown to bind to and to inhibit ABCB- and PIN-mediated auxin transport[6,50–54]. Nearly identical influx profiles for GDA and NPA (Fig. 3a; Supplementary Fig. 5a, d) further supported the concept that both inhibitors interfere with the same targets, most likely ABCBs[50]. This is indirectly reinforced by the finding that influx was similarly reduced in *abcb1 abcb19* roots,

although to a slightly lower degree (47% inhibition; ref. 50). Also, in *hsp90.1* (33%) and *hsp90.3* (39%) loss-of-function roots, auxin transport was significantly and likewise reduced but still sensitive to GDA and NPA (Fig. 3a; Supplementary Fig. 5b–d) supporting further functional redundancy between these HSP90 isoforms.

Indirect electrochemical quantifications of auxin transport were corroborated by the classical application of radiolabeled auxin (IAA) tracers to roots followed by scintillation counting of root segments. GDA significantly inhibited shootward (basipetal) and rootward (acropetal) IAA transport in Arabidopsis roots (Fig. 3b). The HSP90 inhibitor, radicicol, known to bind to the HSP90 ATP-binding site like GDA but with lower specificity, affected likewise PAT, however, only shootward PAT was inhibited significantly. Basipetal root auxin transport was also significantly and likewise inhibited in *hsp90.1*, *hsp90.3*, and both *HSP90^RNAi* lines (the latter though with higher variability) but not in *hsp90.2* and *hsp90.4* (Fig. 3c). Again, strong root PAT reductions (roughly 50–80% inhibition) between *hsp90*, *abcb* and *twd1* mutants argued for action in the same pathway (Fig. 3c). Notably, in all experiments, no significant changes in transport of the diffusion control, benzoic acid (BA), were observed (Supplementary Fig. 5e, f).

Finally, to further investigate the impact of GDA on ABCB-mediated BL transport[12–14], we quantified ATP-dependent ³H-BL uptake into microsomes prepared from root-enriched Wt and *abcb1 abcb19* mutants treated with GDA. Interestingly, GDA in Wt significantly reduced BL transport to levels similar to *abcb1 abcb19*. However, *abcb1 abcb19* BL transport was still sensitive to GDA indicating the presence of other GDA-sensitive BL transporting systems in the absence of ABCB1 and ABCB19 (Supplementary Fig. 5i).

Next, we examined the effect of genetic or pharmacological HSP90 downregulation on root auxin responses by quantifying the activity of the degradation-based auxin reporter, *DII:VENUS*[55]. In agreement with PAT defects and previous analyses using the *DR5-GUS* reporter[28], genetic or pharmacological HSP90 inhibition reduced auxin responses in the root tip as indicated by enhanced activity of DII-Venus (Fig. 3e, f). Reduction of auxin signaling by GDA in the Wt or *HSP90^RNAi* lines was in the same range as found for *abcb1 abcb19* and *twd1*[50,56]. These data were substantiated by quantification of free IAA in 9 dag Arabidopsis seedlings by GC-MS. In agreement with strongly reduced shootward PAT found for HSP90 depletion caused by GDA (Fig. 3b) or genetics (Fig. 3c), we found elevated auxin concentrations

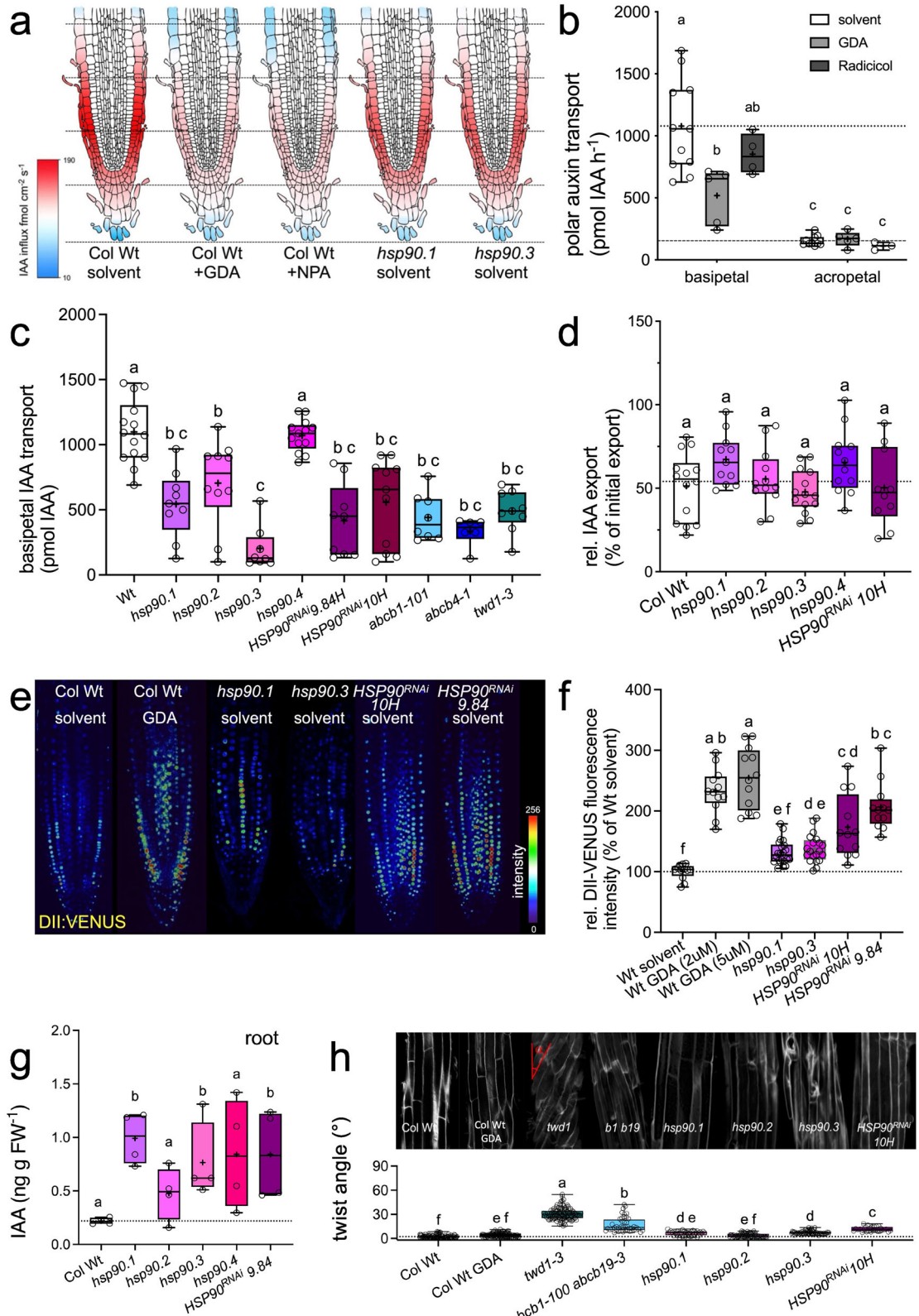

in *hsp90.1*, *hsp90.3*, *HSP90^RNAi* and *twd1-3* roots but not in *hsp90.2* and *hsp90.4* (Fig. 3g). Reduced auxin signaling caused by HSP90 depletion might seem at first to conflict with enhanced root concentrations of IAA. However, it is important to recall that signaling was quantified in the root tip, while free IAA was quantified over the entire root, and IAA signaling is only an indirect read-out for endogenous auxin levels. Moreover, shoot-ward and root-ward PAT attributed to ABCB1 and

ABCB19, respectively[5,17,56], are affected differently by GDA (see Fig. 3b), which correlates with our findings that ABCB1 and ABCB19 own different sensitivities (Fig. 4a). Finally, despite our finding that the triple *tir1afb2afb3* mutants did not show altered root elongation upon GDA treatment (Supplementary Fig. 6h), we cannot entirely exclude the impact of HSP90 depletion on TIR1/AFB protein destabilization due to HSP90 depletion[27]. Auxin concentrations in the shoots of *hsp90*

**Fig. 3 | HSP90 acts as a positive regulator of root ABCB-mediated polar auxin transport. a** Epidermal IAA influx profiles along solvent or indicated drug-treated (each 5 µM) Arabidopsis wild-type and *hsp90* roots. Positive fluxes represent a net IAA influx; $n = 3$. Heat-map presentation of epidermal influx rates; dashed lines indicate 100-µm distances from the root tip. Raw data in Supplementary Fig. 5. **b** Root basipetal but not acropetal PAT is inhibited by Geldanamycin (5 µM) treatment of 5 µM seedlings. Significant differences ($p < 0.05$) of means ± SE ($n = 6$ independent experiments, each with 10 seedlings) were determined using Ordinary One-way ANOVA (Tukey's multiple comparison test) and are indicated by different lowercase letters; benzoic acid (diffusion) control in Supplementary Fig. 5. **c** Root basipetal PAT is inhibited in Arabidopsis *hsp90* loss-of-function and *HSP90^RNAi* mutant lines. Significant differences ($p < 0.05$) of means ± SE ($n = 3$ independent experiments, each with 5 seedlings) were determined using Ordinary One-way ANOVA (Dunnett's multiple comparisons test) and are indicated by different lowercase letters; benzoic acid (diffusion) control in Supplementary Fig. 5. **d** Mesophyll protoplast IAA transport is not significantly affected in Arabidopsis *hsp90* loss-of-function and *HSP90^RNAi* mutant lines. Significant differences ($p < 0.05$) of means ± SE ($n = 10$–$12$ independent protoplast preparations) were determined using Ordinary One-way ANOVA (Tukey's multiple comparison test) and are

indicated by different lowercase letters; benzoic acid (diffusion) control in Supplementary Fig. 5. **e, f** Relative auxin responses are altered by pharmacological or genetical inhibition of HSP90 visualized (**e**) and quantified (**f**) using the degradation auxin reporter, DII-VENUS. Significant differences ($p < 0.05$) of means ± SE ($n = 3$ independent experiments with each 5 seedlings) were determined using Ordinary One-way ANOVA (Tukey's multiple comparison test) and are indicated by different lowercase letters. **g** Root auxin (IAA) levels quantified by GC-MS are increased in *hsp90.1*, *hsp90.3*, and *HSP90^RNAi* mutant roots but not in *hsp90.2* and *hsp90.4*. Significant differences ($p < 0.05$) of means ± SE ($n = 4$ independent experiments, each with 10 roots) were determined using Ordinary One-way ANOVA (Dunnett's multiple comparisons test) and are indicated by different lowercase letters; shoot data in Supplementary Fig. 5. **h** Epidermal root twisting (twist angle) is significantly increased in *twd1*, *hsp90.1*, *hsp90.3*, and *HSP90^RNAi* mutant lines. Significant differences ($p < 0.05$) of means ± SE ($n = 4$ independent experiments, each with 10 roots) were determined using Ordinary One-way ANOVA (Dunnett's multiple comparisons test) and are indicated by different lowercase letters. Data are presented as box-and-whisker plots, where median and 25th and 75th percentiles are represented by the box itself and the middle line, respectively; means are indicated by a "+". Source data are provided as a Source data file.

mutant seedlings (Supplementary Fig. 5h) and auxin transport capacities for *hsp90* mutant mesophyll protoplasts were not significantly affected (Fig. 3d, Supplementary Fig. 5g) arguing that HSP90-controlled auxin transport is profound in roots.

The described PAT defects caused by pharmacological HSP90 inhibition can be expected to have a major impact on physiological and developmental cues[8]. As a first parameter of auxin-controlled growth processes, we examined the effect of GDA on root elongation rates. As reported[28], GDA at low concentrations had no significant effect on Wt root elongation ($IC_{50} = 155$ µM, $R^2 = 0.004$; Supplementary Fig. 6a) and only a mild but not significant effect on *hsp90* roots (Supplementary Fig. 6f). Interestingly, the elongation of *twd1* roots was hypersensitive ($IC_{50} = 198.7$ nM, $R^2 = 0.884$) to GDA (Supplementary Fig. 6a, e), which could be reverted by expression of *35S:HA-TWD1* and *TWD1:TWD1-CFP* (Supplementary Fig. 6e). The HSP90 ATPase inhibitors radicicol ($IC_{50} = 48.5$ nM, $R^2 = 0.751$) and (-)-epigallocatechin gallate ($IC_{50} = 3.8$ µM, $R^2 = 0.719$), both binding to the HSP90 C-terminus, blocked *twd1* root elongation similarly (Supplementary Fig. 6b, c), while novobiocin, shown to act as a low-affinity, unspecific competitive inhibitor of the C-terminal ATPase pocket[27], was far less effective ($IC_{50} = 17.4$ µM, $R^2 = 0.799$) and had a similar effect on *twd1* as on Wt (Supplementary Fig. 6d). Interestingly, root elongation of all tested mutants of the *PIN* and AUX1/LAX families of auxin transporters were not differently affected by GDA compared to Wt (Supplementary Fig. 6g). Importantly, the impact of HSP90 inhibition by GDA was independent of transcriptional auxin responses described recently[27] or brassinosteroid signaling, another hormone known to control plant growth. Root elongation was likewise sensitive to GDA in the triple F-box co-receptor mutants, *tir1/afb1/afb3* and *tir1/afb2/afb3*[57,58], and in brassinosteroid receptor mutants, *bri1* and *bak1*[59] as in Wt (Supplementary Fig. 6h).

A hallmark of the "*twisted dwarf1* syndrome" is the non-handed, helical rotation of epidermal layers of around 30° that is thought to be caused by unequal elongation of epidermal and cortical cell files most likely due to PAT defects[60]. Epidermal twisting was found to be significantly increased by about a factor 2 in seedlings grown on GDA as well as by a factor 3 in single *hsp90.1* and *hsp90.3* but not in *hsp90.2* mutants (Fig. 3h). Interestingly, root twisting in the *HSP90^RNAi* line 10H (11.7° ± 0.6) was significantly enhanced in comparison to the Wt (2.4° ± 0.2) and comparable to *abcb1 abcb19* (13.2° ± 1.0).

These datasets provide clear evidence that cytosolic HSP90 proteins, HSP90.1 and HSP90.3, act as positive regulators of auxin transport and have a profound impact on PAT. Notably, HSP90 action is temporarily, spatially, and mechanistically distinct from a recently

described nuclear module in that HSP90 was reported to stabilize the nuclear auxin receptor, TIR1 in response to ambient temperature elevation[27]. Importantly, while the nuclear presence of HSP90 appears to be required for the transcriptional integration of auxin signaling and the perception of environmental cues[27], ABCB destabilization by HSP90 inhibition is observed under standard conditions, suggesting that HSP90-mediated ABCB stabilization may be constitutive rather than stress-related.

## HSP90 provides plasticity to ABCB plasma membrane presence and auxin-controlled plant development

While analyzing GDA sensitivities of different ABCBs (Fig. 2), we found that the PM presence of all three ABCBs that were shown to be co-chaperoned by TWD1 and HSP90 clients[1,2] is affected by GDA but with different sensitivities. A thorough time-dependent analysis (Supplementary Fig. 7a) revealed that GDA reduced the levels of all three ABCBs at the PM of cells of the root tip (Fig. 4a, b) and of the root transition zone (Supplementary Fig. 7c) in the order ABCB1 ≫ ABCB19 > ABCB4. An identical behavior was found for both *HSP90^RNAi* lines (Fig. 4c, d; Supplementary Fig. 7e).

Interestingly, this differential HSP90-dependent behavior of ABCB1, ABCB19, and ABCB4 seems to match their suggested mobilities during PM trafficking[61]. To verify this correlation, we quantified the sensitivities of these three ABCBs toward the trafficking inhibitor Brefeldin A (BFA). BFA reduces PM labeling and induces protein accumulation of members of all three major auxin transporter classes in so-called BFA compartments[62,63]. A time- and concentration-dependent investigation using quantification of "BFA body area" as a read-out, uncovered remarkably similar BFA sensitivities expressed by an increase of empirical $EC_{50}$ values in the order ABCB1 (2.13 µM) ≫ ABCB19 (3.46 µM) > ABCB4 (4.74 µM) ≅ PIN2 (7.67 µM) (Fig. 4e, f, Supplementary Fig. 7f–i). We also quantified "BFA body number" under the same concentration (50 µM, 1 h) indicating similar BFA sensitivities for the ABCBs resulting in the same order (ABCB1 ≅ PIN2 > ABCB19 > ABCB4), except for PIN2, which in this case is more similar to ABCB1 (Supplementary Fig. 7h). As mentioned above, also ConcA and Wort affected ABCB sorting and trafficking towards lytic vacuoles (Supplementary Fig. 2j).

To test the direct impact of TWD1 on the cycling turnover and the client status of ABCBs, we quantified BFA and GDA sensitivities of ABCB1 in *twd1* complemented with Wt and helix 7 mutant versions of TWD1. We found that mutational uncoupling of HSP90 binding to TWD1 did not significantly influence ABCB1 BFA sensitivities (Supplementary Fig. 1i, j) indicating that HSP90 either does not alter ABCB1

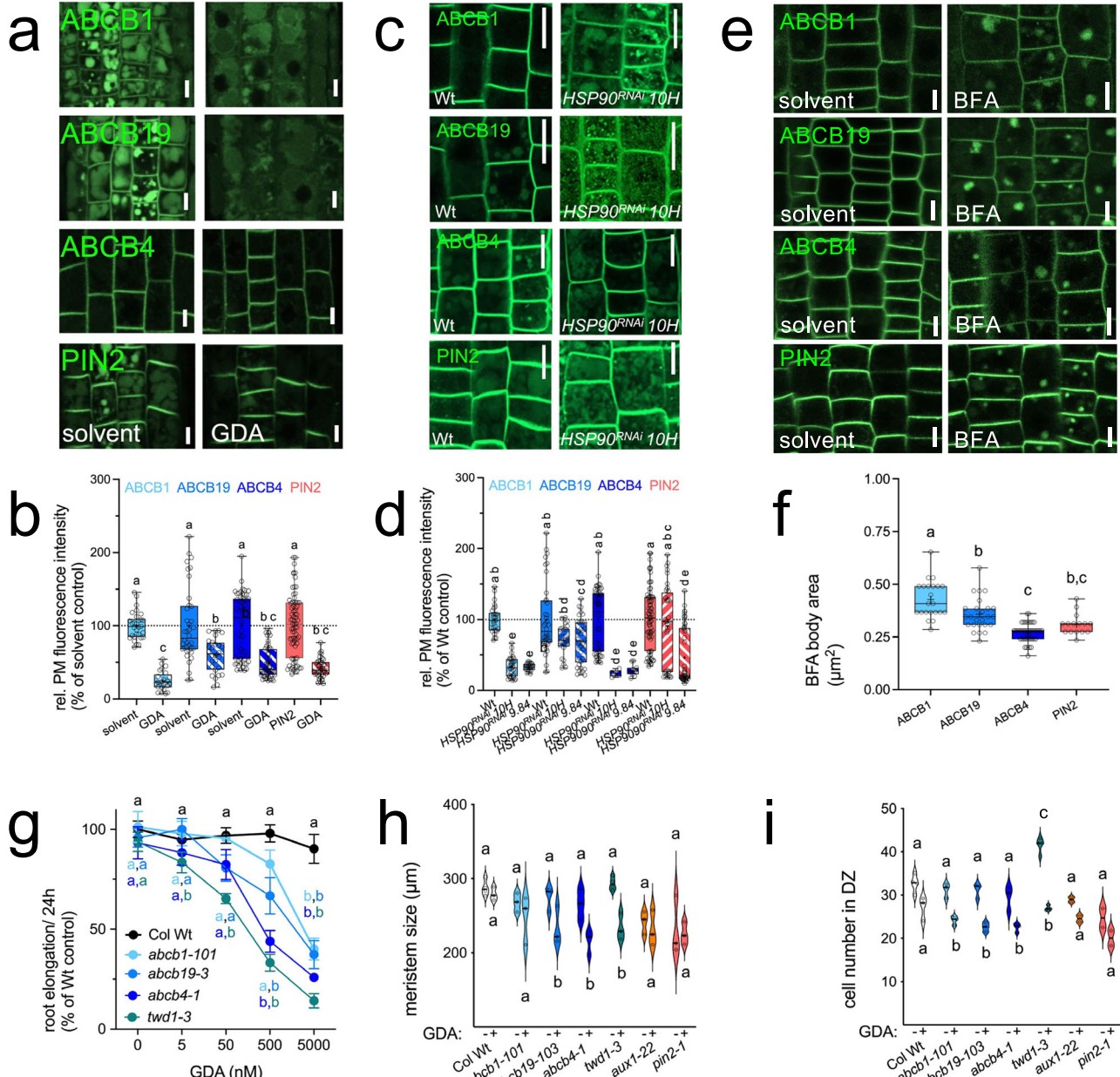

**Fig. 4 | HSP90 provides plasticity to ABCB plasma membrane presence and auxin-controlled plant development. a, b** GDA treatment (5 μM) differentially destabilizes root tip ABCBs. Confocal imaging (**a**) and quantification of PM fluorescence signals in the root tip, corresponding to PM-specific analysis of manually traced ROIs precisely outlining the plasma membrane to extract membrane-localized signal was performed using the Zen Blue 2012 software package (**b**). Significant differences (*p* < 0.05) of means ± SE (*n* = 3 independent GDA treatments with 25–30 cells) were determined using Ordinary One-way ANOVA (Šídák's multiple comparisons test) and are indicated by different lowercase letters; bars, 25 μm. **c, d** *HSP90^RNAi* differentially destabilizes ABCBs in the root tip. Confocal imaging of *HSP90^RNAi* (line 10H) (**c**) and quantification of PM fluorescence signals in root tips of HSP90^RNAi lines *10H* and *9.84*, corresponding to PM-specific analysis of manually traced ROIs precisely outlining the plasma membrane to extract membrane-localized signal was performed using the Zen Blue 2012 software package (**d**). Significant differences (*p* < 0.05) of means ± SE (*n* = 4 independent experiments with 25–30 cells) were determined using Ordinary One-way ANOVA (Šídák's multiple comparisons test) and are indicated by different lowercase letters; bars, 100 μm. **e, f** ABCBs own different sensitivities to Brefeldin A (BFA, 50 μM) in

the root tip. Confocal imaging (**e**) and quantification of BFA body areas in the root tip (**f**). Significant differences (*p* < 0.05) of means ± SE (*n* = 3 BFA treatments with 20–30 cells) were determined using Ordinary One-way ANOVA (Tukey's multiple comparisons test) and are indicated by different lowercase letters; bars = 50 μm. **g** Root elongation of abcb and twd1 mutants is differentially inhibited by GDA treatments (5 μM) in the order Wt ≪ *abcb1* < *abcb19* < *abcb4* < *twd1*. Significant differences (*p* < 0.05) of means ± SE (*n* = 3 independent experiments, each with 10 roots) were determined using Ordinary One-way ANOVA (Šídák's multiple comparisons test) and are indicated by different lowercase letters. **h, i** Meristem size (**h**) and cell number (**i**) of *abcb* and *twd1* mutants is differentially inhibited by GDA treatments (5 μM) in the order Wt ≪ *abcb1* < *abcb19* < *abcb4* < *twd1*. Significant differences (*p* < 0.05) of means ± SE (*n* = 3 independent experiments with each 10 roots) were determined using Ordinary One-way ANOVA (Tukey's multiple comparisons test) and are indicated by different lowercase letters. Data are presented as box-and-whisker plots, where median and 25th and 75th percentiles are represented by the box itself and the middle line, respectively; means are indicated by a "+". Source data are provided as a Source data file.

cycling or that ABCB1 cycling is already in saturation or could not be further accelerated by ABCB1 destabilization due to HSP90 uncoupling. The latter option is supported by the finding that ABCB1 and PIN2 have similar BFA sensitivities based on the BFA body number as a read-out (Supplementary Fig. 7). In contrast, the helix 7 mutation TWD1[F326K] significantly reduced the GDA-sensitivity of ABCB1 in comparison to Wt (Fig. 1i, j), which further underlines that TWD1 determines the HSP90 client status of ABCBs.

Next, to test if this striking correlation between gradual ABCB PM destabilization by HSP90 depletion and ABCB PM recycling speed for these three ABCBs had its expected impact on plant physiology, we quantified root elongation and gravitropic bending, two hallmarks of auxin-controlled physiology and development[64]. Single *abcb* loss-of-function mutants showed an increasing GDA sensitivity of root elongation in the order *abcb1* ($IC_{50} = 3.0\,\mu M$, $R^2 = 0.833$) < *abcb19* ($IC_{50} = 2.2\,\mu M$, $R^2 = 0.579$) ≪ *abcb4* ($IC_{50} = 0.6\,\mu M$, $R^2 = 0.833$; Fig. 4g). This precise phenomenological inverse behavior uncovers an elusive HSP90 mechanistic and gradual stabilization of ABCB isoforms on the PM, revealing that removal of more stable ABCB isoforms has a greater impact on plant physiology. In line with this concept, we found that root elongation in *twd1*, which can be seen as a triple *ABCB1,19,4* loss-of-function mutant[1,2], is even more severely affected by GDA ($IC_{50} = 0.2\,\mu M$, $R^2 = 0.873$; Fig. 4g). While this gradual GDA sensitivity of single *abcb* mutants (Fig. 4; Supplementary Figs. 7, 8) is intrinsically logical, it still raises the question why GDA sensitivity increases in *twd1*, in the apparent PM absence of all three ABCB clients of HSP90 action. An answer to this might come from the finding that brassinosteroid transport and signaling, also known to contribute to cell elongation, seems to be controlled as well by HSP90[24,25]. Besides the recent report of ABCB1 and ABCB19 to export brassinosteroids[12–14], TWD1 was shown to be essential for PM stabilization[25] and trafficking[65,66] of the brassinosteroid receptor, BRASSINOSTEROID INSENSITIVE1. Our results, thus, pinpoint towards a conjunctional auxin-brassinosteroid crosstalk controlled by HSP90, which eventually enforces developmental plasticity.

Lately, HSP90 inhibition was reported to alter root bending to the gravitropic vector[27,28], which is the consequence of auxin-mediated unilateral root elongation[8,67]. We confirmed these data and found that gravitropism of *abcb1*, *abcb4*, and *abcb19* roots showed in principle an identical, hypersensitive performance on GDA (Supplementary Fig. 8) as was also found for root elongation (Fig. 4g). Remarkably, whereas the sensitivity of *twd1* exceeded all three tested *abcb* mutants, *pin2*, and *aux1* were only mildly affected similarly to Wt, indicating that GDA interfered rather with auxin transport-mediated cell elongation than with gravity perception.

To get a deeper understanding of the underlying cellular mechanisms of how HSP90 depletion blocks ABCB-mediated root elongation and bending, we quantified meristem sizes, cell numbers, and cell sizes in the root division zone. Confocal microscopy analyses revealed that both meristem size and the number of cells in division zone cell files in *abcb* and *twd1*, but not in *aux1* and *pin2* or WT, were sensitive to GDA, thus contributing to reduced root growth (Fig. 4h, i). An unexpected and previously overseen result was that *twd1* roots reveal, despite being overall shorter[17], a greatly enhanced cell number in the division zone that is reduced to Wt levels by GDA (Fig. 4i). Analysis of cell sizes revealed that in *twd1* roots, the meristematic division zone cells are indeed smaller than in Wt (Supplementary Fig. 8a, b, f, g), resembling *abcb1 abcb19* roots[12]. However, HSP90 inhibition by GDA resulted in altered cell-size distributions in *abcb* and *twd1* mutant root meristems, whereas Wt, *pin2*, and *aux1* meristems were practically unaffected (Supplementary Fig. 8). In addition, to the more apical onset of cell elongation in *twd1* (Supplementary Fig. 9), GDA treatment resulted in the appearance of enlarged cells throughout the DZ of *twd1* and *abcb* mutants, but not in *aux1* and *pin2* roots. These results suggest that inhibition of HSP90 may unbalance auxin-dependent cell division and cell elongation processes by interfering with ABCB-mediated auxin transport, resulting in altered cell size profiles along meristems of transporter mutants affected differently in PAT.

Together, these findings show that pharmacological or genetic HSP90 depletion significantly affects all tested parameters of auxin-controlled root physiology and development in an action that depends on ABCBs. GDA sensitivities of ABCB1,4,19 PM presence correlate with altered growth phenotypes in *abcb* mutants (Fig. 4), suggesting that HSP90 buffers ABCB auxin transport capacities, providing plasticity to auxin-controlled development.

## Discussion

In conclusion, we identify cytoplasmic HSP90 chaperones as a major player of auxin and most likely also of brassinosteroid transport regulation by recruiting a subset of ABCB-type auxin exporters as a plant kingdom-specific class of HSP90 client proteins. HSP90 gradually stabilizes mature ABCBs at the PM and prevents vacuolar degradation (Fig. 2; Supplementary Fig. 7). Importantly, HSP90 seems to provide a buffer by accommodating steady state levels of ABCB1,4,19 transport capacities at the PM. The strict correlation between ABCB cycling turnover and HSP90-mediated PM stabilization suggests that HSP90 adjusts ABCB-mediated auxin transport activities and auxin-controlled developmental plasticity. Interestingly, our data uncover that HSP90 does not seem to have an impact on ABCB PM cycling as shown by BFA treatment of ABCB1 (Supplementary Fig. 1i, j) but that ABCB cycling speed has an impact on their GDA sensitivity (Fig. 4).

Several lines of evidence presented here suggest that ABCB-HSP90 interaction is only relevant once the ABCBs arrive at the PM. Therefore, an urgent question is whether TWD1 and HSP90 are jointly recruited to the ABCBs on the PM or if the TWD1-ABCB complex recruits HSP90s once it arrives at the PM. To test these hypotheses, we thoroughly reanalyzed PM signal intensities of all three ABCBs and TWD1 in the absence and presence of GDA in the root transition and calculated the ratios of those. GDA treatment resulted in similar ratios for the ABCBs but not for TWD1, which was significant different (Supplementary Fig. 7k), concluding that ABCBs and TWD1 independently interact with HSP90 during translocation to PM. If either of the two needed firstly to associate with HSP90 to mediate the PM translocation of the second, then both would have had very similar PM signal ratios upon GDA treatment. Likewise, if TWD1 interacts first with the ABCBs and then with HSP90, again their PM signal ratios would be expected to be very similar.

Recently, HSP90s were suggested to regulate PIN1 polarity and thus plant physiology[28]. In addition, we here show that GDA significantly alters PIN2 PM presence (Fig. 4) but only mildly *pin2* physiology (Fig. 4; Supplementary Figs. 7, 8, 9). In agreement, we were unable to demonstrate PIN1-HSP90.3 interaction (Fig. 1a), which is a requirement for an HSP90 client status. However, we cannot exclude PIN1 physical interaction with other cytoplasmic HSP90 members. Non-cytoplasmic HSP90 has been recently reported to regulate PIN1 and PIN5 protein abundance[68], which was also shown by *HSP90* downregulation[28], while in the present work we support that PAT defects due to ABCB PM destabilization can indirectly alter PIN1 polarity. PIN1 polarity and abundance were shown to depend on local auxin concentrations[69]. Finally, PIN and ABCB function are connected genetically and physically[70,71], which might suggest that described effects on PIN1 polarity might be indirect. In summary, it seems that at least some PINs depend on HSP90, but it is still an open question if PINs are also HSP90 clients.

Like for HERG and CFTR/ABCC7 regulation by FKBP38[18,20,72,73], ABCB PM trafficking and maturation is shared between HSP90 and its co-chaperone, FKBP42/TWD1. While HSP90 is thought to act indirectly on CFTR folding by inhibiting the PPIase activity of FKBP38[22], here we identify a direct role of HSP90s in PM stabilization of a subset of ABCB

transporters. Interestingly, such a regulatory mechanism has not been described for non-auxin-transporting ABCBs or mammalian ABCBs. Specific interaction between TWD1 and ABCB1,19,4 and PM stabilization of the exact three ABCBs by HSP90 suggests that the client status of these ABCBs is directed by TWD1-HSP90 interaction providing specificity. This concept offers therefore an elegant explanation why mammalian ABCBs are apparently not HSP90 clients; to our knowledge no FKBP38-ABCB interaction has yet been described. Still, a cross-kingdom conservation of such a regulatory ABC-HSP90-FKBP module is somewhat surprising as CFTR/ABCC7 and ABCB1, 19, 4 belong to different ABC subclasses. Strikingly, both ATAs and CFTR are the only ABC transporters that export anions, IAA⁻ and Cl⁻, respectively[6,9,74,75]. Future work will show if the functional ABC-HSP90-FKBP modules for plant ABCBs and mammalian CFTR/ABCC7 function in analogy and have evolved independently.

## Methods

### Plant material and phenotypic analyses

The following *Arabidopsis thaliana* lines in ecotype Wassilewskija (Ws) were used: *twd1-1* (At3g21640[17]); *TWD1:TWD1-CFP*[76]; *ABCB1:ABCB1-GFP*, *ABCB19:ABCB19-GFP*[77] and *hsp90.7/shepherd* (At4g24190[78]). *Pin2/eir1-4* (At5g57090[79]; *twd1-3*[17]; *abcb1-3*[78]; *abcb1-100 abcb19-3*[76]; *ABCB4:ABCB4-GFP* (At2g47000[80]; *35S:HA-TWD1*[17]; *PIN2:PIN2-*GFP[81]; *DII:VENUS*[55], *hsp90.1* (At5g52640; SALK_007614), *hsp90.2* (At5g56030; SALK_038646), *hsp90.3* (At5g56010; SALK_013240*)* and *hsp90.4* (At5g56000; SALK_084059) were all in the Columbia Wt (Col-0). To generate the *HSP90:HSP90.1-neonGFP* cassette, genomic DNA fragments of full-length *HSP90.1* (At5g52640) without stop codon, including the promoter region of 2 kb were amplified from Col-0 genomic DNA and cloned at the BamHI/XbaI sites of the modified pCambia1301 vector to get a *neonGreen* fusion. The *HSP90:HSP90.1-neonGFP* construct was transformed into the *Agrobacterium tumefaciens* strain GV3101 and Col-0 Wt Arabidopsis plants were transformed using the floral dip method.

*DII:VENUS* and *HSP90:HSP90.1-neonGFP* lines were crossed into *twd1-3*, *hsp90* mutants and *HSP90*[RNAi 28] lines or *TWD1:TWD1-CFP*, respectively, and isogenic, homozygous lines for the transgene in the F3 generations were used for further analyses.

Seedlings were generally grown on vertical plates containing 0.5 Murashige and Skoog media, 1% sucrose, and 0.75% phytoagar in the dark or at 8 h (short day), 16 h (long day), or 24 h (constant) light per day. Developmental parameters, such as root gravitropism, root elongation, meristem size, and cell numbers were quantified by microscopy as described in Wang et al. (2013)[1]. Inhibitor treatments were performed for 24 h using between 0 (DMSO) to 5 μM geldanamycin or NPA. In some cases, GDA treatment was performed in the dark in order to enhance vacuolar degradation. Root orientation of epidermal layers to the growth direction (twist angle) was quantified microscopically using agarose imprints as described elsewhere[1]. All experiments were performed at least in triplicate with 30 to 40 seedlings per experiment.

### Protein–protein interaction analyses

For HSP90.1 and HSP90.3 co-immunoprecipitation, *35S:RFP-HSP90.1* or *35S:RFP-HSP90.3* were co-transfected with *35S:TWD1-RFP*, *35S:ABCB1-YFP*, *35S:AUX1-YFP*, or *35S:RFP* with in tobacco was carried out as described in ref. 29. IP was performed using anti-GFP or anti-RFP μMACS columns (Miltenyi Biotec). For Western detection anti-RFP (Red Fluorescent Protein; mouse monoclonal; Agrisera Product no: AS15 3033) or anti-GFP (Rabbit polyclonal antibody to Green Fluorescent Protein, ChromoTek Cat No. pabg1) was used at 1:000 dilution. Equal presence of baits was controlled by Western analyses.

For FRET-FLIM analysis were performed as described elsewhere[42]. In short, indicated binary vectors and *p19* as gene-silencing suppressor were transformed into *Agrobacterium tumefaciens* strain GV3101 and

infiltrated into *Nicotiana benthamiana* leaves. The measurements were performed 3 dai using a SP8 laser scanning microscope (Leica Microsystems) as described[42].

For BRET analysis, *N. benthamiana* leaves were Agrobacterium co-infiltrated with indicated BRET construct combinations (or corresponding empty vector controls) and microsomal fractions were prepared 4 days after inoculation (dai). BRET signals were recorded from microsomes (each ~10 μg) in the presence of 5 μM coelenterazine (Biotium Corp.) using the Cytation 5 image reader (BioTek Instruments) and BRET ratios were calculated as described previously[1]. The results are the average of 20 readings collected every 30 s, presented as average values from a minimum of three independent experiments (biological replica: independent Agrobacterium infiltrations and microsome preparations) each with four technical replicates.

### Quantitative proteomics using TMT-based multiplexing

For quantitative proteomics, Columbia Wt (Col-0) or indicated Arabidopsis *HSP90* or *TWD1* mutant seedlings were grown in mixotrophic liquid cultures (½ MS, 1% sucrose) for 12 days. In some cases, cultures were treated with 5 μM geldanamycin for 24 h. Total microsomes were prepared and proteins were extracted and TMTpro labeling was performed as described elsewhere[42]. LC-MS/MS measurements were performed on an Exploris 480 mass spectrometer coupled to an EasyLC 1200 nanoflow-HPLC (all Thermo Scientific) as described in ref. 42. MS raw files were analyzed using ProteomeDiscoverer (version 2.5, Thermo Scientific) using a Uniprot full-length *A. thaliana* database and common contaminants. Exported normalized abundances for each channel were further analyzed using Perseus software. GO-term analyses were performed with Cytoscape 3.8.2 and ClueGO 2.5.9.

### Auxin transport

Simultaneous ³H-indolyl-3-acetic acid (IAA; ARC ART0340, 25 Ci/mmol) and ¹⁴C-benzoic acid (BA; ARC ART0186A, 55 mCi/mmol) export from *Arabidopsis* and tobacco mesophyll protoplasts was analyzed as described[35]. Equal protoplast loading was achieved by substrate diffusion into protoplasts on ice, and export was started by a temperature shift (25 °C). Relative export from protoplasts was calculated from exported radioactivity into the supernatant as follows: (radioactivity in the supernatant at time t = x min) − (radioactivity in the supernatant at time t = 0)) * (100%)/(radioactivity in the supernatant at t = 0 min); presented are mean values from >4 independent protoplast preparations. ³H-brassinolide (BL) uptake, custom-synthesized at ARC (20 Ci/mmol) into *Arabidopsis* root-enriched vesicles prepared from Arabidopsis lines grown in the dark liquid cultures was measured as described elsewhere[82].

Root basipetal and acropetal PAT measurements simultaneous allowing to quantify ³H-IAA and ¹⁴C-BA transport were performed as described in ref. 83. A platinum microelectrode was used to monitor IAA fluxes in *Arabidopsis* roots as described previously (Mancuso et al., 2005). For measurements, Col Wt plants or *hsp90* mutants were grown in hydroponic cultures and used at 5 dag. Differential current from an IAA-selective microelectrode was recorded in the absence and presence of 5 μM NPA or geldanamycin.

### In planta analysis of auxin contents and responses

Endogenous free IAA were quantified from shoot and root segments of 9 dag *Arabidopsis* seedlings by gas chromatography-mass spectrometry (GC-MS) as described in ref. 84. Methylation was performed by adding equal sample amounts of a 1:10 diluted solution (in diethylether) of trimethylsilyldiazomethane solution (Sigma-Aldrich) for 30 min at room temperature. The mixture was then evaporated and resuspended in 50 ml of ethyl acetate for GC-MS analysis. Data are means of four independent lots of 30–50 seedlings each, and equivalent to ca. 100 mg root shoot material, respectively.

Homozygous generations of *Arabidopsis twd1-3* expressing *DII:VENUS* were obtained by crossing with *DII:VENUS* lines[55]. For analyses of *DII:VENUS* responses, seedlings were grown vertically for 5 dag and for 24 h on 5 μM GDA plates and analyzed by confocal laser-scanning microscopy.

## Confocal microscopy

5 dag wild type and *HSP9ORNAi* seedlings expressing *ABCB1:ABCB1-GFP*, *ABCB19:ABCB19-GFP*, *ABCB4:ABCB4-GFP*, *PIN2:PIN2-GFP*, *AUX1 AUX1-YFP*, *TWD1:TWD1-YFP* and *DII:VENUS* were using a Zeiss LSM 800 equipped with Airyscan laser scanning confocal microscope (Carl Zeiss, Germany). GFP was detected with 488 nm excitation laser line (2.00% laser intensity, 488–536 nm collection bandwidth, 1.00 gain level), YFP was detected with 514 nm excitation laser line (2.00% laser intensity, 520–550 nm collection bandwidth, 1.00 gain level), CFP was detected with 405 nm excitation laser line (2.00% laser intensity, 475–485 nm collection bandwidth, 1.00 gain level), and VENUS was detected with 515 nm excitation laser line (2.00% laser intensity, 515–540 nm collection bandwidth, 1.00 gain level). Images were produced and post-acquisition processed using Zeiss ZEN 2012 Blue software (Carl Zeiss, Oberkochen, Germany). Total fluorescence intensity was quantified by measuring average signal intensity across entire root regions (either meristematic or transition zones). For plasma membrane (PM)-specific measurements, we carefully delineated regions of interest (ROIs) following the PM contour and recorded the mean fluorescence intensity within these boundaries.

## Statistics and reproducibility

Data were statistically analyzed using Prism 10.1.1. (GraphPad Software, San Diego, CA). Normal (Gaussian) distribution of values was tested prior to statistical analyses using the Prism-embedded D'Agostino-Pearson omnibus normality test. For statistics, either "Ordinary One-way ANOVA" or "Two-way ANOVA" and indicated comparison tests were used as post hoc analysis to determine significances. Data are presented as "box-and-whisker plots", where median and 25th and 75th percentiles are represented by the box itself and the middle line; means are indicated by a "+".

No statistical method was used to predetermine sample size. No data were excluded from the analyses; the experiments were not randomized; the investigators were not blinded to allocation during experiments and outcome assessment.

## Reporting summary

Further information on research design is available in the Nature Portfolio Reporting Summary linked to this article.

## Data availability

Requests for data should be made to and will be fulfilled by M.M. Geisler (markus.geisler@unifr.ch) and P. Hatzopoulos (phat@aua.gr), provided the data will be used within the scope of the originally provided informed consent. Source data are provided with this paper.

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

## Acknowledgements

We would like to thank L. Charrier and S. Rößler for excellent technical assistance, F. l'Haridon, S. Robert for *bri1* and *bak1* and *tir1/afb1/afb3* seeds, and M. Estelle for *afb2* mutant seed combinations, respectively. This work was supported by grants from the Hellenic Foundation of Research and Innovation (project HFRI-2020-02746 to P.H.), the Czech Science Foundation (GA 20-20860Y to M.Z. and V.P.) and the project TowArds Next GENeration Crops (reg. no. CZ.02.01.01/00/22_008/0004581 of the ERDF Programme Johannes Amos Comenius to T.N.) and the Swiss National Funds (project 31003A_165877 and 310030_197563 to M.G.).

## Author contributions

M.M.G., A.B., T.T. and M.d.D. designed research; T.T., M.d.D., D.S., V.P., D.M., E.A., M.Z., F.R.I., K.P. and P.K.P. performed research; T.T., M.d.D., D.S., D.M., E.A., S.M., M.Z., T.N., M.S., J.L.M., P.H. and M.M.G. analyzed data; S.M., T.N., J.L.M., P.H. and M.M.G. supervised work; M.M.G. wrote the manuscript, all authors commented on the manuscript.

## Competing interests

The authors declare no competing interests.
