## [Transparent Peer Review file · Nature Communications]

HSP90 differentially stabilizes plant ABCB-type auxin transporters on the plasma membrane

Corresponding Author: Dr Markus Geisler

Version 0:

Reviewer comments:

Reviewer #1

(Remarks to the Author)

The manuscript by Tsering et al entitled "HSP90 provides plasticity to plant development by gradually stabilizing plasma membrane presence of ABCB-type auxin transporters" assess the function of HSP90 in stabilizing ABCGB transporters via TWD1 at the PM. While the manuscript offers intriguing and novel insights, it is not yet ready for publication, as key experiments are still too preliminary and lack essential controls.

Major points:

- The key messages in the results section are sometimes unclear. I recommend improving the structure by including titles for the individual subchapters.
- Additionally, some parts of the results section would be more appropriate in the discussion, as they are too speculative. For example, lines 179-181: The "small but significant upregulation of TWD1 on the ER". This needs to be demonstrated experimentally. The single CLSM image shown in Figure 2C rather suggests the opposite. Lines 184-188: What evidence supports the suggestion that "trafficking defects might be mediated by a different HSP90 isoform that controls the ESCRT pathway"? The citation provided here is also inappropriate.
- In general, the figure legends should provide more detailed information about the experiments, including the sample size ("n"), which is often missing or vaguely described as ">3." For instance, in line 499, it simply says "Quantification," and in line 514, "Quantification of total signal," leaving the reader unclear about the methods used for quantification. Additionally, key abbreviations should be explained, such as the plant line abbreviation "b1-100 b19-3" in Figure 3h. Moreover, a corresponding image for the ColO wild-type plant line with GDA is missing in the upper panel, and the term "twisti angle" should be corrected to "twist angle."
- Experiments from Figure 1 are key experiments for this paper and therefore need to be more convincing and with all appropriate controls.
- For Figure 1a, the anti-GFP Western blot should clearly show both the TWD1-YFP as well as the YFP control band in the total input section. I also recommend repeating the co-immunoprecipitation (Co-IP) experiment with the TWD1F326K and TWD1N187A or TWD1K265A to reconfirm the FRET-FLIM results, which alone are not very convincing. This is a key experiment on which the manuscript is based, so the binding must be demonstrated more convincingly.
- Figure 1b: This experiment requires appropriate controls. Representative CLSM images should be provided to confirm that all constructs are expressed at similar levels. Additionally, the sample size for the different FRET-FLIM pairs is not indicated. A larger sample size would likely help ensure that the observed differences are highly statistically significant.
- Figure 1c: The authors claim that the mutated versions of TWD1 do not significantly impact TWD1's ability to promote ABCB1-mediated auxin export. However, how do they reconcile this statement with the finding that there is no significant difference between the control and TWD1F326K?
- Extended Data Figure 1b: The figure legend states that co-expression does not alter ABCB1 location in tobacco epidermal leaves. I disagree with this claim because it is nearly impossible to draw meaningful conclusions about localization or changes in localization from transient expression in tobacco leaves. Additionally, the authors use the same experiment (line

128) to evaluate alterations in expression, for which transient expression in tobacco leaves is also unsuitable.

- Figure 1i: Please include CLSM images taken using the same settings, allowing the reader to evaluate the loss of fluorescence in the *twd1* background. Additionally, could you clarify what is being measured as "fluorescence intensity" in Figure 1j, and how large the sample size (n) is? Conducting the same experiment with TWD1F326K and TWD1N187A expressing plant line in the ABCB1:ABCB1-GFP in *twd1* background would significantly strengthen the conclusion. Alternatively, membrane fractionation experiments followed by immunoblotting could be used to confirm the loss of PM localization of ABCB1-GFP.

- Figure 2 including extended data:

- o Figure 2e is missing a negative control.

- o If the authors want to claim that there is a shift to the tonoplast (line 167-168), they need to provide better pictures than 2a or Supplemental Figure 2e and try to assess a statistical difference between the two lines.

- Lastly, for the BFA treatment: the recommended approach to assess alterations in recycling is to count the number of BFA bodies rather than their area. This is because the size of BFA bodies can vary significantly depending on the duration of the treatment. Specifically, for ABCB19, there is considerable vacuolar staining (as shown in Fig 4a and Supplementary Figure 6A). Therefore, to ensure accurate analysis, co-treatment with FM4-64 and BFA is necessary as only BFA bodies co-stain. Please clarify which parameters are measured in terms of relative fluorescence intensity in Fig. 4b,d and Supplementary Figure 6b-e.

- Minor points:

- o Line 57: (PAT,2,3,7...second parentheses is missing

- o Line 69: AUX1LAX AUX1/LAX

Reviewer #2

(Remarks to the Author)

In this study, Tsering et al. explore the dynamic roles of HSP90 and its co-chaperone TWD1 in stabilizing plasma membrane ABCB-type auxin transporters, influencing polar auxin transport and conferring developmental plasticity in plants. The data presented is robust and compelling. However, I have several suggestions to enhance the clarity and depth of the findings:

Major Points:

1. The manuscript would benefit from details on the expression patterns of TWD1 and HSP90 at both tissue and subcellular levels. Can native TWD1 co-localize with HSP90 in root cells, considering their complex function in regulating ABCB-type auxin transporters? Although Figure 1a and 1b demonstrate FRET-FLIM assays in transiently transformed tobacco leaves, concerns remain regarding potential differences in spatio-temporal expression patterns between TWD1 and HSP90.

2. Additional evidence is necessary to assess the influence of the helix 7 mutation TWD1F326K on the interaction with HSP90.3, beyond the FRET-FLIM assays shown in Figure 1b. This would provide a more comprehensive understanding of the role of F326 in this interaction.

3. It would be insightful to explore whether vacuolar degradation or endocytosis pathways are involved in the degradation of PM-localized ABCB1/19 induced by pharmacological or genetic depletion of HSP90. Could inhibitors specific to these pathways restore ABCB1/19 PM stability?

4. The manuscript often lacks information on replicates for cytological quantifications. It would improve the reliability of the data if this detail were included in the figures or figure legends.

5. The relationship between reduced root cell PAT due to HSP90 depletion (as shown in Figure 3f,g) and the increased levels of free auxin (Figure 3g), despite a decrease in auxin signaling (Figure 3f), needs further clarification.

6. Given the recent identification of ABCB19 as a brassinosteroid transporter, it would be interesting to investigate whether brassinosteroid distribution in roots can be modulated by GDA treatment, as discussed in Lines 376-381.

Minor Points:

1. There is an incomplete sentence at Line 36 that needs to be addressed.

2. The Results section could be improved with the addition of sub-titles to enhance readability and structure.

3. In Figure 1a, the YFP band appears to be inappropriately cropped in the elution channel. Please ensure that the full band is displayed.

4. The absence of a negative control in Figure 2e is concerning. Additionally, it would be beneficial to specify the number of replicates conducted for the Co-IP assay.

5. Figure 1h lacks corresponding quantifications of the division pattern, which would aid in the interpretation of the data.

6. In Figure 1j, quantifying the intensity of ABCB1 PM in a *twd1* mutant background would provide valuable insights.
7. For Figures 2a and 2c, please specify which promoters were used to drive the expression of the indicated genes.
8. The X-axis in Extended Data Figure 2d is unclear. Clarification of what it represents would aid in data interpretation.
9. The manuscript mentions data related to dark treatment in Lines 172-174, but this data appears to be missing or not clearly presented.

Reviewer #3

(Remarks to the Author)

The manuscript by Tsering and co-workers identifies ABCB1,19,4 as clients for HSP90 proteins. This interaction is necessary for stabilizing specifically these ABCB proteins at the plasma membrane, and thus determine the auxin transport potential. In line with this the HSP90 inhibitors such GDA, act as auxin transport inhibitors. The activity of these ABCBs is famously regulated by TWD1, impacting on its ER export and transport activities at the PM. Here, HSP90 were identified as TWD1 interactors. A mutation that disturbs TWD1-HSP90 interaction largely rescues the *twd1* phenotype, indicating that the TWD1-HSP90 interaction is not essential for ABCB ER export. Overall, the authors make a compelling case supported by a variety of biochemical and genetic data. I do have some comments that would need to be addressed.

- The title is not very accurate, and would need rephrasing. The developmental plasticity effect does not really emerge from the data, as the HSP90 seem to act constitutively, and at least one of the target ABCBs does not uniquely transport auxin.
- Despite the central role of HSP90, its position in the introduction is a bit minimal. From the introduction, it should also be clear that HSP90 are known to regulate auxin and JA signaling. Similarly, the fact that ABCB19 was shown to transport, besides IAA, also another plant hormone should be mentioned already in the introduction.
- A lot of data is presented showing that TWD1 interacts with HSPs, and the importance of both proteins on ABCB activity and protein stability is demonstrated. Yet, it remains not well described what is the exact interrelationship between these proteins in stabilizing ABCB on the PM. As they are co-chaperones, both are essential. They interact with each other, and with ABCBs. But their interaction is not necessary for ABCB ER exit. It thus seems that this interaction is thus only relevant once the ABCBs arrived at the PM. Are they jointly recruited to the PM-ABCB, or does the TWD1-ABCB complex recruit HSPs once it arrived at the PM. More details of the role of TWD1-HSP90 in the HSP90-ABCB interaction, eg. using FRET-FLIM in WT and *twd1* backgrounds, are needed.
- GDA is well known as an inhibitor of HSP90 activity, probably via inhibition of the ADP/ATP binding pocket of the protein. This affect HSP90-ABCB or HSP90-TWD1 interaction?
- The HSP90 interaction is proposed to be constitutive under the conditions tested. How does this fit with an important role in developmental plasticity via tuning ABCB activities. The recent finding that brassinosteroids are transported via ABCBs and that BL reduces ABCB levels, make me wonder if the transport substrates IAA and BL modify ABCB PM residence and the HSP90-ABCB interaction? In the same context, it would be nice to test for BL related effects in GDA treated plants, and *hsp90* mutants.
- In a caption, it is mentioned that the GDA effect is strictly root specific. While this is of interest, there is no data provided to support this claim. And if this is the case, what does this mean for all assays (transport and interaction) involving the transient expression in *N benthamiana*?
- Line 47: what is meant by extracellular auxin maxima? As far as I understand it, auxin maxima/minima have mostly been inferred from transcriptional auxin responses and measurements of auxin contents in individual cells. I have no knowledge of the maxima being restricted to the apoplast.
- Line 69: AUX1LAX -> AUX1/LAX
- Line 143: the statement of "essential for correct cell division" related to the TWD1-HSP90 interaction is somewhat inflated, as the majority of cells seem quite normal. In any case, this statement needs to be supported by quantifications.
- Line 151: states HSP90 as cytoplasmic proteins. Is this demonstrated in literature? If so, introduce this in the introduction.
- Line 184-187: please rephrase this sentence for more clarity.
- Line 209: microscopical data. Gives the impression of very small data, rather than data generated using a microscope
- Line 281-287 discusses the discrepancy between measured auxin content and auxin response marker activities. I found it strange that at this point no mention is made of the reduction of TIR1/AFB mediated auxin signaling as a possible explanation. To exclude this option, it would be important to demonstrate that in their conditions GDA does not affect TIR1/AFB protein stability before dismissing this possibility, as was done a bit later based on *tir1afb1afb3*. The choice of this auxin signaling mutant is odd, as the *afb1* has a minor effect on long term auxin responses. A better choice would have been

combinations with *afb2*.

- BFA treatment is interesting to study the trafficking dynamics of PM proteins. It should also be noted that BFA also has rapid effect on PM peripheral proteins such as D6PK. That ABCB-HSP90 interaction is proposed to be mostly relevant for ABCB proteins at the PM, one can imagine that BFA could impact directly or indirectly on the interaction. In such a scenario, effects on GDA could no longer be detected. An alternative method for estimating trafficking dynamics should be implemented.

- Previous reports demonstrate complex genetic interactions between *pin* and *abcb* mutants, and effects of ABCB on PIN protein stability. Yet, here *pin2* mutants responded largely as WT to GDA. Moreover, GDA does not affect PIN trafficking. How should this be understood/explained.

- Avoid using **** for indicating significance. *** is more than enough

- Define what is n in the legends. Is this individual experiments, either as a single experimental unit and/or independent biological repeats.

- In Fig 2i, there is no ABCB19. Yet this seems one of the more important ABCBs for this manuscript.

Version 1:

Reviewer comments:

Reviewer #1

(Remarks to the Author)

Thank you for the thorough revisions and detailed responses. I am satisfied with the changes made and recommend the paper for publication

Reviewer #2

(Remarks to the Author)

I appreciate the authors' and comprehensive revision of the manuscript and their detailed responses to my prior comments. Overall, the authors have made a substantial effort to address the concerns raised, and the revised version represents an improvement in both clarity and depth. I have no further comments.

Reviewer #3

(Remarks to the Author)

The authors have answered all my comments and suggestions either experimentally or in text. Beside the details below, I have no additional comments to this manuscript

Line 106: NeonGFP -> mNeonGreen

Line 232: hardened -> corroborated/strengthened

Line 284-285 seems to suggest that NPA only targets ABCBs. Yet, structural data have shown NPA binding to PINs. Rephrase; both inhibitors share/converge on a common target

Line 347: AUX1 -> italics

Line 386-390 describes K_m and V_{max} for BFA sensitivities of trafficking of ABCB1/19/4 and PIN2. I am not convinced that this can be easily calculated. BFA targets ARF-GEFs with specific kinetics and K_m , V_{max} make sense to me in that context, but not really in BFA body size context. What does a V_{max} expressed at $\mu\text{mol}/\text{min}$ even mean? The corresponding trend lines in Fig. S7 g and i are suggestive, but no data points are gathered outside the saturated part of the fitted curve...

Line 468. As the data also showed effects on BL transport, selectivity for auxin seems to be too restrictive

Fig. 3a. While aesthetically pleasing, the overlay of IAA uptake in the different zones is somewhat inaccurate, as it suggests that anatomy is not changed between the different mutants and treatments and that auxin uptake in the inner layers is uniform. Probably, such color coding would be more suited along a Y-axis that has its origin at the very tip of the root meristem? A cartoon to roughly indicate what the distances likely refer to, can be kept next to such a graph.

Suppl Fig S1 contains 2x panel c. I suppose the c in the TWD1 panel should be deleted.

The use of exactly same color for the text in Figs as the fluorescent signal make it sometimes difficult to read exactly what is written. Eg. Suppl Fig. S1f

Suppl Fig S2a. HSP90 levels seem to go up when treated with CHX. This is unexpected.

Supple Fig S2 legend. Cycloheximid -> cycloheximide

Suppl Fig S4a and e. Labels for IAA, IAA+GDA, BL and BL+GDA are shifted to the right (BL+GDA reads as BL)

Check legends for consistent use of n or N

Reviewer #1 (Remarks to the Author)

The manuscript by Tsering et al entitled "HSP90 provides plasticity to plant development by gradually stabilizing plasma membrane presence of ABCB-type auxin transporters" assess the function of HSP90 in stabilizing ABCGB transporters via TWD1 at the PM. While the manuscript offers intriguing and novel insights, it is not yet ready for publication, as key experiments are still too preliminary and lack essential controls.

We would like to thank Reviewer 1 for his/her overall positive evaluation and various suggestions that allowed us to improve the quality of our manuscript.

Major points:

R1.1: The key messages in the results section are sometimes unclear. I recommend improving the structure by including titles for the individual subchapters.

As suggested, we have now added titles to the subchapters.

R1.2: Additionally, some parts of the results section would be more appropriate in the discussion, as they are too speculative. For example, lines 179-181: The "small but significant upregulation of TWD1 on the ER". This needs to be demonstrated experimentally. The single CLSM image shown in Figure 2C rather suggests the opposite.

We have removed all speculative parts of the Results section and moved some parts to the Discussion when appropriate. Concerning our statement on Fig. 2c, we have removed the "ER" from the text as the quantification allows to judge for increased total signals, which we think is also reflected by the corresponding Figure 2C.

R1.3: Lines 184-188: What evidence supports the suggestion that "trafficking defects might be mediated by a different HSP90 isoform that controls the ESCRT pathway"?

We fully agree and have removed the speculative part on an involvement of the ESCRT pathway.

R1.4: The citation provided here is also inappropriate.

The inappropriate citation was removed with the entire part making an exchange obsolete.

R1.5: In general, the figure legends should provide more detailed information about the experiments, including the sample size ("n"), which is often missing or vaguely described as ">3." For instance, in line 499, it simply says "Quantification," and in line 514, "Quantification of total signal," leaving the reader unclear about the methods used

for quantification. Additionally, key abbreviations should be explained, such as the plant line abbreviation "b1-100 b19-3" in Figure 3h. Moreover, a corresponding image for the ColO wild-type plant line with GDA is missing in the upper panel, and the term "twisti angle" should be corrected to "twist angle."

We fully agree and have now improved the information given in the legends in respect to correct the sample sizes. The method for image quantification was added to the Method section and abbreviations were clarified.

Concerning Fig. 3h, The Wt GDA picture was added, and the misspelling was corrected.

R1.6: Experiments from **Figure 1** are key experiments for this paper and therefore need to be more convincing and with all appropriate controls.

As explained below, we have now added the requested controls.

R1.7: For Figure 1a, the anti-GFP Western blot should clearly show both the TWD1-YFP as well as the YFP control band in the total input section.

We have repeated the entire co-IP as requested under R1.8 but exchanged the tags (TWD1 is now RFP-tagged, while HSP90 carries a YFP, like for the FRET assay). As a negative control, we now use free RFP. As requested, the Western blot of new Fig. 1b shows a clear signal for both baits, RFP and TWD1-RFP.

R1.8: I also recommend repeating the co-immunoprecipitation (Co-IP) experiment with the TWD1F326K and TWD1N187A or TWD1K265A to reconfirm the FRET-FLIM results, which alone are not very convincing. This is a key experiment on which the manuscript is based, so the binding must be demonstrated more convincingly.

As suggested, we have indeed repeated the co-IP with TWD1^{F326K} and TWD1^{K265A} and HSP90.3, respectively, which is now shown as new Fig. 1b. The results overall support the conclusion from FRET-FLIM experiments showing reduced TWD1-HSP90.3 interaction for the helix7 mutation (TWD1^{F326K}), while TWD1^{K265A} shows similar interaction as Wt TWD1.

The discrepancy between FRET-FLIM and co-IP results likely reflects their distinct sensitivity to molecular interactions. While FRET detects nanometer-scale distance changes that can be significantly affected by point mutations, co-IP relies on larger interaction interfaces (protein domains) despite single amino acid substitutions. We have incorporated this interpretation in the Results section.

In order to have consistent datasets, we also included now TWD1^{K265A} into Figs. 1e-j.

R1.9: Figure 1b: This experiment requires appropriate controls. Representative CLSM images should be provided to confirm that all constructs are expressed at similar levels. Additionally, the sample size for the different FRET-FLIM pairs is not indicated. A larger sample size would likely help ensure that the observed differences are highly statistically significant.

We have included in the current version confocal images of co-expression experiments forming the basis for FRET-FLIM experiments of Fig. 1b as Extended Fig. 1f indicating that TWD1 and HSP90.3 are co-expressed to similar levels. We have also increased and adjusted the sample size of FRET-FLIM measurements, which is now also indicated in the figure legend.

R1.10: Figure 1c: The authors claim that the mutated versions of TWD1 do not significantly impact TWD1's ability to promote ABCB1-mediated auxin export. However, how do they reconcile this statement with the finding that there is no significant difference between the control and TWD1F326K?

We would like to remind the reviewer that these data are generated in tobacco, owning an endogenous TWD1 ortholog that most likely provides biogenesis and PM stabilization of the Arabidopsis ABCB1.

We believe that slightly reduced (though non-significant) transport rates were caused by unequal sample sizes. Therefore, we have repeated transport measurements resulting now in equal sample sizes that clearly show that the difference between ABCB1 and TWD1^{F326K} is indeed also statistically different. Our data argue that HSP90 binding to helix 7 has no direct effect on the up-regulation of ABCB1-mediated auxin transport by TWD1, which is thought to be provided by the PPlase of the TWD1 FKBD (Geisler and Hegedus 2020; Geisler et al. 2016).

R1.11: Extended Data Figure 1b: The figure legend states that co-expression does not alter ABCB1 location in tobacco epidermal leaves. I disagree with this claim because it is nearly impossible to draw meaningful conclusions about localization or changes in localization from transient expression in tobacco leaves. Additionally, the authors use the same experiment (line 128) to evaluate alterations in expression, for which transient expression in tobacco leaves is also unsuitable.

We agree that in many (but not all) cases it is indeed difficult to draw safe conclusions of localization based on images of epidermal layers, although we strongly believe that at least the signals for ABCB1 in Extended Fig. 1b are of very good quality and most likely are of PM origin. However, what was actually meant here (and similarly for other figures) is that co-expression of Wt or mutated TWD1 does not alter significantly expression levels of ABCB1 (and vice versa). This is corrected throughout the legends and text. Further, we have quantified co-localization of ABCB1 and TWD1, which is now added as Extended Fig. 1e.

R1.12: Figure 1i: Please include CLSM images taken using the same settings, allowing the reader to evaluate the loss of fluorescence in the twd1 background. Additionally, could you clarify what is being measured as "fluorescence intensity" in Figure 1j, and how large the sample size (n) is? Conducting the same experiment with TWD1F326K and TWD1N187A expressing plant line in the ABCB1:ABCB1-GFP in twd1 background would significantly strengthen the conclusion. Alternatively, membrane fractionation experiments followed by immunoblotting could be used to confirm the loss of PM localization of ABCB1-GFP.

As requested, we provide new confocal images taken under the same conditions and included *TWD1*^{K265A}. As before, additional data reveal that the helix 7 mutation (*TWD1*^{F326K}) reduced the ABCB1 PM presence, which was also found for the TPR mutation (*TWD1*^{K265A}), although to less extent.

R1.13: Figure 2 including extended data: Figure 2e is missing a negative control.

A negative control is included showing that *AUX1*-YFP is unable to pull-down *HSP90.1* and *HSP90.3*, as expected (revised Fig. 2e).

R1.14: If the authors want to claim that there is a shift to the tonoplast (line 167-168), they need to provide better pictures than 2a or Supplemental Figure2e and try to assess a statistical difference between the two lines.

We have thoroughly discussed this issue and feel that the vacuolar delocalisation of ABCB1 in +GDA treatment is well supported by high quality pictures that are now presented in Extended Fig. 2e. We would like to point out that these data are nicely complemented by our Western analyses after sucrose gradient centrifugation that shows a clear vacuolar shift for ABCB1 but not for PIP.

We add-below for review a plot of the signal intensity through the vacuoles. Signal plot profiles indicate that the GFP signal is accumulating inside the vacuolar compartment (mock), whereas with GDA there is less GFP signal in vacuoles and the GFP plot pattern is similar to the FM4-64, indicating that the localization of the ABCB1 is rather on the tonoplast.

MS dark 24h FM4-64[4] 3h

GDA[5]24h FM4-64[4] 3h

Finally, we have, as requested, performed a quantification of vacuolar ABCB1-GFP fluorescence for data presented in Extended Fig. 2e, showing a clear reduction of vacuolar signals for GDA treatment in comparison to the solvent control; the quantification is now added as Extended Fig. 2f.

R1.15: Lastly, for the BFA treatment: the recommended approach to assess alterations in recycling is to count the number of BFA bodies rather than their area. This is because the size of bodies can vary significantly depending on the duration of the treatment. Specifically, for ABCB19, there is considerable vacuolar staining (as shown in Fig 4a and Supplementary Figure 6A). Therefore, to ensure accurate analysis, co-treatment with FM4-64 and BFA is necessary as only BFA bodies co-stain. Please clarify which parameters are measured in terms of relative fluorescence intensity in Fig.4b,d and Supplementary Figure 6b-e.

We discussed this point and consulted similar publications. We still believe that the “BFA body area” is indeed a more appropriate measure in comparison with the “BFA body number” as the area can change between tested samples (see Fig. 4). Anyway, we also quantified the BFA body number under the same concentration (50 μ M, 1h) and results ($ABCB1 \cong PIN2 > ABCB19 > ABCB4$) in principle correspond to the “BFA body areas” ($ABCB1 \gg ABCB19 > ABCB4 \cong PIN2$), with the exception of PIN2 that is in BFA body number more similar to ABCB1. These data are now included as Extended Figure 7j.

As outlined under comment R3.13, we have also employed Wortmannin and Concomycin A treatments. Conclusively, Concomycin A and Wortmannin inhibitors stabilized and restored the PM localization of ABCB1 and ABCB19, while all three inhibitors (GDA, Conca and Wort) affected their sorting and trafficking towards lytic vacuoles.

Parameters used for the quantification of fluorescence signals are now indicated in the Methods and in the figure legends.

Minor points:

Line 57: (PAT,2,3,7...second parentheses is missing.

The parenthesis is now added.

Line 69: AUX1LAX à AUX1/LAX

This has now been corrected, thanks for pointing this out.

Reviewer #2 (Remarks to the Author):

In this study, Tsering et al. explore the dynamic roles of HSP90 and its co-chaperone TWD1 in stabilizing plasma membrane ABCB-type auxin transporters, influencing polar auxin transport and conferring developmental plasticity in plants. The data presented is robust and compelling. However, I have several suggestions to enhance the clarity and depth of the findings:

We would like to thank Reviewer 2 for his/her overall positive evaluation and his/her various suggestions that allowed us to improve the quality and clarity of our manuscript.

Major Points:

R2.1: The manuscript would benefit from details on the expression patterns of TWD1 and HSP90 at both tissue and subcellular levels. Can native TWD1 co-localize with HSP90 in root cells, considering their complex function in regulating ABCB-type auxin transporters? Although Figure 1a and 1b demonstrate FRET-FLIM assays in transiently transformed tobacco leaves, concerns remain regarding potential differences in spatio-temporal expression patterns between TWD1 and HSP90.

This is(was) indeed a shortcoming that we have now addressed as follows: First, we have added in silico root expression taken from the eFP browser (<http://bbc.botany.utoronto.ca/efp/cgi-bin/efpWeb.cgi>) that verify an overlapping expression pattern for TWD1 and involved HSP90 isoforms (and ABCB1) in the root tip. Secondly, we generated HSP90.1:HSP90.1-mNeonGreen lines and crossed them with TWD1:TWD1-CFP lines and imaging indicates that both proteins do indeed co-express in the root transition zone; in silico and imaging data were added as new Extended Fig. 1a and 1b, respectively.

R2.2: Additional evidence is necessary to assess the influence of the helix 7 mutation TWD1F326K on the interaction with HSP90.3, beyond the FRET-FLIM assays shown in Figure 1b. This would provide a more comprehensive understanding of the role of F326 in this interaction.

As suggested also by Reviewer 1, we have repeated the co-IP with TWD1^{F326K} and TWD1^{N187A} and HSP90.3, which is now shown as new Fig. 1b. The results support the conclusion from FRET-FLIM experiments showing reduced TWD1-HSP90.3 interaction for the helix7 mutation (TWD1^{F326K}). The discrepancy between these two techniques might be caused by the fact that point mutation can lead to altered distances between proteins that are detected by FRET, while substitution of single amino acids might have less effect on co-IPs as protein-protein interactions employ entire surface domains; this is now added to the Results sections.

R2.3: It would be insightful to explore whether vacuolar degradation or endocytosis pathways are involved in the degradation of PM-localized ABCB1/19 induced by pharmacological or genetic depletion of HSP90. Could inhibitors specific to these pathways restore ABCB1/19 PM stability?

We used Concanamycin A (ConcA), a specific inhibitor of the vacuolar H⁺-ATPases to reduce the acidification of lytic compartments and thus protein degradation⁴⁰. Under our experimental conditions, we noticed different sensitivities of the ABCBs to ConcA with ABCB19 and ABCB1 being the most sensitive ones. (Extended Fig. 2j) and responding with increased abundance at the PM upon ConcA treatment indicating

aberrant endomembrane trafficking (Extended Fig. 2j). Contrary, ConcA decreased the presence of ABCB4 at the PM (Extended Fig. 2j). Notably, ConcA application in GDA-treated seedlings did not change the PM levels of ABCB1 and ABCB19 when compared to GDA-treated plants (Extended Fig. 2j). Next, we used wortmannin (Wort), a specific phosphatidylinositol 3-kinase (PI3K) inhibitor, which inhibits plasma membrane protein and receptor sorting and/or vesicle budding required for the delivery of endocytosed material to "mixing" endosomes ⁴¹.

Wort induced the internalization of ABCB1 and ABCB19 showing impaired vacuolar trafficking indicating that ABCB1 and ABCB19 require PI3K signaling for their translocation to lytic vacuoles, while ABCB4 was found to be insensitive to Wort treatment (Extended Fig. 2j). We observed mistargeting of ABCB1 and ABCB19 to the tonoplast suggesting sorting defects at the level of multivesicular body (MVB)/PVC (Extended Fig. 2j). In GDA-treated plants, WT seems to inhibit early stages of endocytosis of all tested ABCB transporters at the plasma membrane (Extended Fig. 2j).

Conclusively, all three inhibitors (GDA, ConcA and Wort) affected their sorting and trafficking towards lytic vacuoles of ABCB1 and ABCB19 and had a minor effect on the ABCB4.

R2.4: The manuscript often lacks information on replicates for cytological quantifications. It would improve the reliability of the data if this detail were included in the figures or figure legends.

We fully agree and apologize for this shortcoming. Now, the number of replicates for each experiment as well as all details on the individual statistical test, is added to the legends of all figures; this had also been requested by Reviewer 3.

R2.5: The relationship between reduced root cell PAT due to HSP90 depletion (as shown in Figure 3f,g) and the increased levels of free auxin (Figure 3g), despite a decrease in auxin signaling (Figure 3f), needs further clarification.

We agree that these two contradictory findings are puzzling at first glance. However, as already explained in the text, we believe that this discrepancy originates from two facts: first, auxin signalling was measured in the root tip, while PAT and auxin quantification were performed over/using the entire root. Second, shoot-ward and root-ward PAT (that were attributed to ABCB1 and ABCB19, respectively (Geisler et al. 2005, Bouchard et al. 2006, etc.) are affected differently by GDA (see Fig. 3b), which correlates with our findings that ABCB1 and ABCB19 own different sensitivities (Fig. 4a). Finally, as pointed out correctly by Reviewer 3 (see R3.12), despite our finding that triple TIR1/AFP mutants did not show altered root elongation with GDA (see Extended Fig. 6h), we cannot entirely exclude that reduction of TIR1/AFB mediated auxin signalling is caused by TIR1/AFB protein destabilization due to HSP90 depletion as shown by the group of Mark Estelle (Wang et al. 2016).

This is now better explained in the Results and the latter options were added.

R2.6: Given the recent identification of ABCB19 as a brassinosteroid transporter, it would be interesting to investigate whether brassinosteroid distribution in roots can be modulated by GDA treatment, as discussed in Lines 376-381.

We agree that the impact of GDA on BR transport by ABCBs is very interesting, but we have left this out because we found that the effect of GDA is much stronger in the root than in the shoot, which did not allow us to assess this by using our leaf protoplast system.

However, to address your question, we have used microsomes from root material (prepared from Wt and abcb1 abcb19 plants grown in liquid cultures in the dark, resulting in “root-like” material) to measure ATP-dependent 3H-BL uptake. Results clearly demonstrate that, as expected, GDA significantly reduces BL transport in the Wt to abcb1 and abcb19 levels. Interestingly, abcb1 abcb19 BL transport was still sensitive to GDA, indicating the presence of other GDA-sensitive BL transporting systems in the absence of ABCB1 and ABCB19; these data are now added as Extended Fig. 5i.

Minor Points:

1. There is an incomplete sentence at Line 36 that needs to be addressed.

This has now been corrected.

2. The Results section could be improved with the addition of sub-titles to enhance readability and structure.

This had been also advised by Reviewer 1 and has been addressed; please see the revised version.

3. In Figure 1a, the YFP band appears to be inappropriately cropped in the elution channel. Please ensure that the full band is displayed.

Upon request of Reviewer 1, we have added an entirely new co-IP that includes also TWD1^{F326K}, which addresses this comment (see new Fig. 1b)

4. The absence of a negative control in Figure 2e is concerning. Additionally, it would be beneficial to specify the number of replicates conducted for the Co-IP assay.

Also, upon request of Reviewer 1, we have added a negative control with AUX1-YFP. All co-IPs were conducted at a minimum in triplicate (3 independent tobacco infiltrations and co-IPs), typical results are presented; this is now added to the Figure legends.

5. Figure 1h lacks corresponding quantifications of the division pattern, which would aid in the interpretation of the data.

*We are not sure what exactly a “quantification of the division pattern” might mean. Is it the “frequency per root” or a similar meaning? In any case, this phenotype is not a new finding for *twd1* and had been already described in earlier work on *ULTRACURVATA2* (*UCU2*) by the Micol group ((Perez-Perez et al. 2004); *UCU2* is identical to *TWD1*), therefore we do not believe that such a quantification would add any further information.*

6. In Figure 1j, quantifying the intensity of ABCB1 PM in a *twd1* mutant background would provide valuable insights.

*Indeed, also upon request of Reviewer 1, we have added images taken under identical confocal settings, and the quantification includes also ABCB1 in *twd1*.*

7. For Figures 2a and 2c, please specify which promoters were used to drive the expression of the indicated genes.

Throughout the imaging (not transport) work, native promoters using well-established lines were used; this is indicated in the Methods and figure legends of the revised version.

8. The X-axis in Extended Data Figure 2d is unclear. Clarification of what it represents would aid in data interpretation.

We apologize for this shortcoming: the “sucrose gradient fraction number” was missing.

9. The manuscript mentions data related to dark treatment in Lines 172-174, but this data appears to be missing or not clearly presented.

This was indeed unclear. In some cases, GDA treatment was performed in the dark in order to enhance PM removal and vacuolar degradation. This information is added to the Methods and figure legend of the revised version.

Reviewer #3 (Remarks to the Author):

The manuscript by Tsering and co-workers identifies ABCB1,19,4 as clients for HSP90 proteins. This interaction is necessary for stabilizing specifically these ABCB proteins at the plasma membrane, and thus determine the auxin transport potential. In line with this the HSP90 inhibitors such GDA, act as auxin transport inhibitors. The activity of these ABCBs is famously regulated by TWD1, impacting on its ER export and

transport activities at the PM. Here, HSP90 were identified as TWD1 interactors. A mutation that disturbs TWD1-HSP90 interaction largely rescues the *twd1* phenotype, indicating that the TWD1-HSP90 interaction is not essential for ABCB ER export. Overall, the authors make a compelling case supported by a variety of biochemical and genetic data. I do have some comments that would need to be addressed.

We would like to thank Reviewer 3 for his/her overall positive evaluation and his various suggestions that allowed us to improve the quality of our manuscript.

R3.1: The title is not very accurate, and would need rephrasing. The developmental plasticity effect does not really emerge from the data, as the HSP90 seem to act constitutively, and at least one of the target ABCBs does not uniquely transport auxin.

We agree that the previous title was a bit speculative and would like to suggest the following that focuses on the key data of our work: "HSP90 differentially stabilizes plant ABCB-type auxin transporters on the plasma membrane".

R3.2: Despite the central role of HSP90, its position in the introduction is a bit minimal. From the introduction, it should also be clear that HSP90 are known to regulate auxin and JA signaling. Similarly, the fact that ABCB19 was shown to transport, besides IAA, also another plant hormone should be mentioned already in the introduction.

As requested, we have added information on the role of HSP90 in hormone signalling and on the transport of BRs by ABCB19 and ABCB1 in the Introduction section of the revised version.

R3.3: A lot of data is presented showing that TWD1 interacts with HSPs, and the importance of both proteins on ABCB activity and protein stability is demonstrated. Yet, it remains not well described what is the exact interrelationship between these proteins in stabilizing ABCB on the PM. As they are co-chaperones, both are essential. They interact with each other, and with ABCBs. But their interaction is not necessary for ABCB ER exit.

It thus seems that this interaction is thus only relevant once the ABCBs arrived at the PM. Are they jointly recruited to the PM-ABCB, or does the TWD1-ABCB complex recruit HSPs once it arrived at the PM. More details of the role of TWD1-HSP90 in the HSP90-ABCB interaction, eg. using FRET-FLIM in WT and *twd1* backgrounds, are needed.

*We fully agree that the role of TWD1 in the ABCB-HSP90 interaction is a central question, therefore, we took your advice to explore individual interaction of these pairs and gave it a deep thought. As suggested, we tried indeed to FRET-FLIM ABCBs in the *twd1* KO but failed due to the very low abundance of the ABCBs (see new Fig. 1i-j).*

As an alternative we came up with the concept that if both ABCB1 and TWD1 interacted independently with HSP90 to move to the PM then we should have different GDA sensitivities or – the other way around - if all 3 came in the same complex on

vesicles from the ER or they form a stable complex on the PM, they should have similar GDA sensitivities.

To test this hypothesis, we carefully reanalyzed PM fluorescence signals of all three ABCBs and TWD1 in the absence and presence of GDA and calculated the ratios of those. Based on the differences in GDA ratios between ABCBs and TWD1, it is safe in our eyes to conclude that both ABCB1 and TWD1 independently interact with HSP90 to move to the PM:

If either of the two components (ABCBs and TWD1) needs first the HSP90 in order for the second component to move to PM, then they both will have a very similar ratio. In the same venue of thinking, if TWD1 interacts first with ABCB1 and then with HSP90, again, the ratio will be very similar. The TWD1 ratio is different from the tested ABCBs, indicating that HSP90s associate independently with TWD1 and ABCBs.

Our argument is also confirmed by Fig.1 i-j, showing that ABCB1 is still present at the PM in *twd1-3*, though at lower levels when compared to the wild type. However, ABCB1 disappears from the PM when *twd1-3* plants are treated with GDA. This means that ABCB1 is GDA sensitive (thus a client) in the absence of TWD1, meaning that TWD1 is not critical for ABCB-HSP90 interaction at the PM, but possibly TWD1 participates in early ABCB biogenesis.

These new data are added now as Extended Fig. 7k and are explained in detail in the Results and Discussion section.

R3.4: The HSP90 interaction is proposed to be constitutive under the conditions tested. How does this fit with an important role in developmental plasticity via tuning ABCB activities. The recent finding that brassinosteroids are transported via ABCBs and that BL reduces ABCB levels make me wonder if the transport substrates IAA and BL modify ABCB PM residence and the HSP90-ABCB interaction? In the same context, it would be nice to test for BL related effects in GDA treated plants, and hsp90 mutants.

Indeed, this is a very important control as many (but not all) ABC transporters are upregulated by their own substrates. Therefore, we decided to quantify PM presence of ABCB1, ABCB4 and ABCB19 in solvent and GDA-treated seedlings and

subsequently applied IAA (1 μ M for 2h) or BL (100 nM for 4h). Primary roots and root transition zones are presented (newly added Extended Figure 4).

Results indicate that all three ABCBs are strongly upregulated by IAA, while the upregulation of ABCB19 and ABCB4 by IAA was stronger in comparison to ABCB1 (Extended Figure 4). Interestingly, only ABCB19 is apparently upregulated by BL; ABCB4 is downregulated, while ABCB1 was widely unaffected by BL (Extended Figure 4). Importantly, hormone treatment had no significant impact on PM protein levels of ABCB1 and ABCB19 when HSP90 function was inhibited by GDA, suggesting that IAA or BL responses require functional HSP90. These findings are now added to the Results section. The discrepancy of ABCB19 up- (our findings) vs. down-regulation in (Ying et al. 2024) is currently under investigation.

R3.5: In a caption, it is mentioned that the GDA effect is strictly root specific. While this is of interest, there is no data provided to support this claim. And if this is the case, what does this mean for all assays (transport and interaction) involving the transient expression in *N benthamiana*?

We apologize for this statement that was based on earlier data indicating that the action of GDA might be root-specific and somehow was left in this footnote. As the effect of GDA is not strictly root-specific, the entire footnote was removed.

R3.6: Line 47: what is meant by extracellular auxin maxima? As far as I understand it, auxin maxima/minima have mostly been inferred from transcriptional auxin responses and measurements of auxin contents in individual cells. I have no knowledge of the maxima being restricted to the apoplast.

Recent findings from different groups (for a recent review, see (Vanneste et al. 2025)) indicate that apoplastic auxin concentrations are perceived by ABP1-like isoforms and transmitted via TMK1 and other receptor kinases to phosphorylate the H⁺-ATPases responsible for fast auxin-responses. This was meant. However, as we feel that these findings still require thorough independent verification, we left the word "apoplastic" out.

R3.7: Line 69: AUX1LAX -> AUX1/LAX

This typo was corrected, thanks.

R3.8: Line 143: the statement of "essential for correct cell division" related to the TWD1-HSP90 interaction is somewhat inflated, as the majority of cells seem quite normal. In any case, this statement needs to be supported by quantifications.

We agree that the role of TWD1 in this process was overinterpreted, as aberrant cell divisions are indeed a rare event, therefore, we have changed the word "essential" to "critical".

R3.9: Line 151: states HSP90 as cytoplasmic proteins. Is this demonstrated in literature? If so, introduce this in the introduction.

It is correct that not all HSP90 proteins are cytoplasmic. Indeed, HSP90.7 is for example found in the ER lumen, while HSP90.5 and HSP90.6 are chloroplastic and mitochondrial. What was meant here is that the HSP90 isoforms that are relevant for this study (because they are TWD1 interacting proteins), like HSP90.1 and HSP90.3 are cytoplasmic, as well as HSP90.2/4, that are used here as negative controls. The cytoplasmic location of these relevant HSP90s is now shown for HSP90.1 in the newly added Extended Fig. 1b.

This has been corrected in the Introduction.

R3.10: Line 184-187: please rephrase this sentence for more clarity.

The sentence has now been simplified as follows: "In contrast to GDA treatment, the reduced PM localization of ABCB1 in both hsp90.1 mutants and HSP90RNAi lines occurred without vacuolar redistribution, implying isoform-specific regulation of ABCB1 trafficking by other HSP90 isoforms."

R3.11: Line 209: microscopical data. Gives the impression of very small data, rather than data generated using a microscope.

Obviously, we meant "cell biological data", which is now used.

R3.12: Line 281-287 discusses the discrepancy between measured auxin content and auxin response marker activities. I found it strange that at this point no mention is made of the reduction of TIR1/AFB mediated auxin signaling as a possible explanation. To exclude this option, it would be important to demonstrate that in their conditions GDA does not affect TIR1/AFB protein stability before dismissing this possibility, as was done a bit later based on tir1afb1afb3. The choice of this auxin signaling mutant is odd, as the afb1 has a minor effect on long term auxin responses. A better choice would have been combinations with afb2.

We are very thankful for this comment as it provides in our eyes another argument explaining the discrepancy between measured auxin contents and auxin response activities; this is now added as a further option to the results although our analyses now include an afb2 mutant combination pointing to the fact that inhibition of root elongation by GDA is at least under our conditions widely independent of TIR1/AFB-mediated auxin signaling.

As requested, the afb2 combination tir1/afb2/afb3 is now added to Extended Fig. 6h.

R3.13: BFA treatment is interesting to study the trafficking dynamics of PM proteins. It should also be noted that BFA also has rapid effect on PM peripheral proteins such as D6PK. That ABCB-HSP90 interaction is proposed to be mostly relevant for ABCB proteins at the PM, one can imagine that BFA could impact directly or indirectly on the

interaction. In such a scenario, effects on GDA could no longer be detected. An alternative method for estimating trafficking dynamics should be implemented.

A similar point was brought up also by Reviewer 2 (R2.3) asking for an alternative method to demonstrate the vacuolar delocalization of ABCB1 caused by BFA treatment.

We used Concanamycin A (ConcA), a specific inhibitor of the vacuolar H⁺-ATPases to reduce the acidification of lytic compartments and thus protein degradation⁴⁰. Under our experimental conditions, we noticed different sensitivities of the ABCBs to ConcA with ABCB19 and ABCB1 being the most sensitive ones. (Extended Fig. 2j) and responding with increased abundance at the PM upon ConcA treatment indicating aberrant endomembrane trafficking (Extended Fig. 2j). Contrary, ConcA decreased the presence of ABCB4 at the PM (Extended Fig. 2j). Notably, ConcA application in GDA-treated seedlings did not change the PM levels of ABCB1 and ABCB19 when compared to GDA-treated plants (Extended Fig. 2j). Next, we used wortmannin (Wort), a specific phosphatidylinositol 3-kinase (PI3K) inhibitor, which inhibits plasma membrane protein and receptor sorting and/or vesicle budding required for the delivery of endocytosed material to "mixing" endosomes⁴¹.

Wort induced the internalization of ABCB1 and ABCB19 showing impaired vacuolar trafficking indicating that ABCB1 and ABCB19 require PI3K signaling for their translocation to lytic vacuoles, while ABCB4 was found to be insensitive to Wort treatment (Extended Fig. 2j). We observed mistargeting of ABCB1 and ABCB19 to the tonoplast suggesting sorting defects at the level of multivesicular body (MVB)/PVC (Extended Fig. 2j). In GDA-treated plants, WT seems to inhibit early stages of endocytosis of all tested ABCB transporters at the plasma membrane (Extended Fig. 2j). Conclusively, all three inhibitors (GDA, ConcA and Wort) affected their sorting and trafficking towards lytic vacuoles of ABCB1 and ABCB19 and had a minor effect on the ABCB4.

R3.14: Previous reports demonstrate complex genetic interactions between pin and abcb mutants, and effects of ABCB on PIN protein stability. Yet, here pin2 mutants responded largely as WT to GDA. Moreover, GDA does not affect PIN trafficking. How should this be understood/explained?

We feel that the genetic interaction is less well-understood, however, the physical and functional interaction between PINs and ABCBs is indeed of great interest. Also, recent modeling work from the Band and Dreyer labs (Geisler and Dreyer 2024; Mellor et al. 2022) supports an interactive action in the widest sense. Therefore, we decided to reanalyze PIN2 GDA sensitivities and crossed PIN2:PIN2-GFP with HSP90 RNAi lines and quantified PM signals. These data are presented in Fig. 4a-c, and indicate that, in contrast to total PIN2 signals (Fig. 2), PIN2 PM localization is indeed mildly affected by GDA treatment and by HSP90 genetic depletion but to a lesser extent than ABCB1. In agreement, also pin2 is more sensitive to GDA than Wt (Extended Fig. 6g), although differences are not significant. As such, we believe that our findings are in overall in agreement with functional PIN-ABCB interactions but that a functional pin2 loss might be partially covered by functionally redundant PIN isoforms (like PIN1) or ABCB1 shown to contribute also to shootward PAT (Bouchard et al. 2006; Geisler et al. 2005; Geisler et al. 2003).

These data suggest that PINs might be eventually also HSP90 clients. This finding, together with functional PIN-ABCB interactions, is now added to the Discussion section.

R3.15: Avoid using **** for indicating significance. *** is more than enough. As requested, we have limited our statistical analyses to a significance of $p < 0.001$.

R3.16: Define what is n in the legends. Is this individual experiments, either as a single experimental unit and/or independent biological repeats.

The identity of “individual experiments” is now explained throughout all legends.

R3.17: In Fig 2i, there is no ABCB19. Yet this seems one of the more important ABCBs for this manuscript.

We have now included also ABCB19 into the BRET analyses (please see the revised Fig.2i) that is in agreement with co-IP data (Zhu et al. 2016) showing an interaction with TWD1.

References

- Bouchard R, Bailly A, Blakeslee JJ, Oehring SC, Vincenzetti V, Lee OR, Paponov I, Palme K, Mancuso S, Murphy AS, Schulz B, Geisler M (2006) Immunophilin-like TWISTED DWARF1 modulates auxin efflux activities of Arabidopsis P-glycoproteins. *J Biol Chem* 281 (41):30603-30612. doi:10.1074/jbc.M604604200
- Geisler M, Bailly A, Ivanchenko M (2016) Master and servant: Regulation of auxin transporters by FKBP and cyclophilins. *Plant Sci* 245:1-10. doi:10.1016/j.plantsci.2015.12.004
- Geisler M, Blakeslee JJ, Bouchard R, Lee OR, Vincenzetti V, Bandyopadhyay A, Titapiwatanakun B, Peer WA, Bailly A, Richards EL, Ejendal KF, Smith AP, Baroux C, Grossniklaus U, Muller A, Hrycyna CA, Dudler R, Murphy AS, Martinoia E (2005) Cellular efflux of auxin catalyzed by the Arabidopsis MDR/PGP transporter AtPGP1. *Plant J* 44 (2):179-194. doi:10.1111/j.1365-313X.2005.02519.x
- Geisler M, Dreyer M (2024) An auxin homeostat allows plant cells to establish and control defined transmembrane auxin gradients. *New Phytologist*. doi:10.1111/nph.20120
- Geisler M, Hegedus T (2020) A twist in the ABC: regulation of ABC transporter trafficking and transport by FK506-binding proteins. *FEBS Lett* 594 (23):3986-4000. doi:10.1002/1873-3468.13983

- Geisler M, Kolukisaoglu HU, Bouchard R, Billion K, Berger J, Saal B, Frangne N, Koncz-Kalman Z, Koncz C, Dudler R, Blakeslee JJ, Murphy AS, Martinoia E, Schulz B (2003) TWISTED DWARF1, a unique plasma membrane-anchored immunophilin-like protein, interacts with Arabidopsis multidrug resistance-like transporters AtPGP1 and AtPGP19. *Mol Biol Cell* 14 (10):4238-4249. doi:10.1091/mbc.E02-10-0698
- Mellor NL, Voss U, Ware A, Janes G, Barrack D, Bishopp A, Bennett MJ, Geisler M, Wells DM, Band LR (2022) Systems approaches reveal that ABCB and PIN proteins mediate co-dependent auxin efflux. *Plant Cell* 34 (6):2309-2327. doi:10.1093/plcell/koac086
- Perez-Perez JM, Ponce MR, Micol JL (2004) The ULTRACURVATA2 gene of Arabidopsis encodes an FK506-binding protein involved in auxin and brassinosteroid signaling. *Plant Physiol* 134 (1):101-117. doi:10.1104/pp.103.032524
134/1/101 [pii]
- Vanneste S, Pei Y, Friml J (2025) Mechanisms of auxin action in plant growth and development. *Nat Rev Mol Cell Biol*. doi:10.1038/s41580-025-00851-2
- Ying W, Wang Y, Wei H, Luo Y, Ma Q, Zhu H, Janssens H, Vukasinovic N, Kvasnica M, Winne JM, Gao Y, Tan S, Friml J, Liu X, Russinova E, Sun L (2024) Structure and function of the Arabidopsis ABC transporter ABCB19 in brassinosteroid export. *Science* 383 (6689):eadj4591. doi:10.1126/science.adj4591
- Zhu J, Bailly A, Zwiewka M, Sovero V, Di Donato M, Ge P, Oehri J, Aryal B, Hao P, Linnert M, Burgardt NI, Lucke C, Weiwad M, Michel M, Weiergraber OH, Pollmann S, Azzarello E, Mancuso S, Ferro N, Fukao Y, Hoffmann C, Wedlich-Soldner R, Friml J, Thomas C, Geisler M (2016) TWISTED DWARF1 Mediates the Action of Auxin Transport Inhibitors on Actin Cytoskeleton Dynamics. *Plant Cell* 28 (4):930-948. doi:10.1105/tpc.15.00726

REVIEWERS' COMMENTS

Reviewer #1 (Remarks to the Author):

Thank you for the thorough revisions and detailed responses. I am satisfied with the changes made and recommend the paper for publication

Reviewer #2 (Remarks to the Author):

I appreciate the authors' and comprehensive revision of the manuscript and their detailed responses to my prior comments. Overall, the authors have made a substantial effort to address the concerns raised, and the revised version represents an improvement in both clarity and depth. I have no further comments.

Reviewer #3 (Remarks to the Author):

The authors have answered all my comments and suggestions either experimentally or in text.

Beside the details below, I have no additional comments to this manuscript.

Thanks to all three reviewers for the overall positive re-evaluation of our work!

Line 106: NeonGFP -> mNeonGreen

mNeonGreen was now used throughout the text and figure legends.

Line 232: hardened -> corroborated/strengthened.

"Corroborated" was now used here.

Line 284-285 seems to suggest that NPA only targets ABCBs. Yet, structural data have shown NPA binding to PINs. Rephrase; both inhibitors share/converge on a common target.

This was indeed not our intention. NPA binding and inhibition of PINs is now integrated and properly referenced.

Line 347: AUX1 -> italics

Corrected.

Line 386-390 describes Km and Vmax for BFA sensitivities of trafficking of ABCB1/19/4 and PIN2. I am not convinced that this can be easily calculated. BFA

targets ARF-GEFs with specific kinetics and K_m , V_{max} make sense to me in that context, but not really in BFA body size context. What does a V_{max} expressed at $\mu\text{mol}/\text{min}$ even mean? The corresponding trend lines in Fig. S7 g and i are suggestive, but no data points are gathered outside the saturated part of the fitted curve...

We discussed this valid point amongst us and consulted experts in the field and tend to agree that a here used Michaelis-Menten (MM) description is indeed not appropriate. The reason is that MM describes an enzyme rate vs. its substrate concentration but that here BFA is an inhibitor of ARF-GEFs, while our readout is BFA body size, a downstream, steady-state morphological consequence of a trafficking blockade, not an initial reaction velocity. Further, BFA body size integrates multiple processes (ARF-GEF inhibition, vesicle budding/fusion, cargo accumulation, etc).

In that respect, we have used now a sigmoidal plot that resulted in a very similar fit but that allowed to calculate empirical EC_{50} values that are an equivalent to K_M 's but obviously not identical. Thanks for tracing this flaw!

Line 468. As the data also showed effects on BL transport, selectivity for auxin seems to be too restrictive.

This is correct. As brassinosteroid transport for ABCBs was described only very recently, most datasets are dealing with a description of the effect of HSP90 on auxin transport. Therefore, we have adjusted our statement as follow, and hope that this is acceptable:

"In conclusion, we identify cytoplasmic HSP90 chaperones as a new major player of auxin and most likely also on brassinosteroid transport regulation by recruiting a subset of ABCB-type auxin exporters as a plant kingdom-specific class of HSP90 client proteins."

Fig. 3a. While aesthetically pleasing, the overlay of IAA uptake in the different zones is somewhat inaccurate, as it suggests that anatomy is not changed between the different mutants and treatments and that auxin uptake in the inner layers is uniform. Probably, such color coding would be more suited along a Y-axis that has its origin at the very tip of the root meristem? A cartoon to roughly indicate what the distances likely refer to, can be kept next to such a graph.

*As explained in our original publication ((Wang et al. 2013); for details, see Fig. 3)), a major limitation of the here used external IAA electrode is that it can only monitor epidermal auxin contents. It is also correct that the here employed heat-map presentation is mainly a visualization of the rather complex influx profiles given in Supplementary Fig. 5. However, we still believe that these graphs are correct and meaningful as the employed HSP90 mutants and short GDA treatments do not have an obvious effect on root morphology and length (which would be obviously different for *twd1* or *abcb1 abcb19* roots). For a better orientation of the distances in relation to the root tip (corresponding to 0), a scale is now added next to the graphs of Fig. 3a and Supplementary Fig. 5d where dashed lines indicate 100- μm distances from the root tip.*

While updating figures, we realized that we somehow during the 1st resubmission had partially pasted-in incorrect versions of the graphs in Suppl. Fig. 5d (GDA and NPA treatments of Col Wt) that were obviously not corresponding to the raw data of Supplementary Fig. 5a-c and also to Fig. 3a. We are very sorry for this. To be entirely transparent with the changes we made, we below attach a comparison of the previous (incorrect) and newly corrected data:

Suppl Fig S1 contains 2x panel c. I suppose the c in the TWD1 panel should be deleted.

The additional “c” was now removed, thanks!

The use of exactly same color for the text in Figs as the fluorescent signal make it sometimes difficult to read exactly what is written. Eg. Suppl Fig. S1f.

We fully understand this issue but tried to label the figures color-wise as intuitive as possible. We have now improved the labels by making them bigger and bold where possible.

Suppl Fig S2a. HSP90 levels seem to go up when treated with CHX. This is unexpected.

We agree that this is unexpected, but CHX might simply cause a general stress to the plant that results in HSP90 upregulation. Please, note also that the used antibody is unable to differentiate between the here relevant HSP90 isoforms, HSP90.1 and HSP90.3, and those that are stress-related, like HSP90.5 and HSP90.7. This is now added to the Results.

Supple Fig S2 legend. Cycloheximid -> cycloheximide.

Corrected, thanks!

Suppl Fig S4a and e. Labels for IAA, IAA+GDA, BL and BL+GDA are shifted to the right (BL+GDA reads as BL)

Thanks for tracing this flaw that is now corrected.

Check legends for consistent use of n or N.

Small "n" are now used throughout the legends and text to indicate the sample size.

References

Wang B, Bailly A, Zwiewka M, Henrichs S, Azzarello E, Mancuso S, Maeshima M, Friml J, Schulz A, Geisler M (2013) Arabidopsis TWISTED DWARF1 functionally interacts with auxin exporter ABCB1 on the root plasma membrane. *Plant Cell* 25 (1):202-214. doi:10.1105/tpc.112.105999